# Stochastic modelling, Bayesian inference, and new in vivo measurements elucidate the debated mtDNA bottleneck mechanism

Iain G Johnston[1], Joerg P Burgstaller[2,3], Vitezslav Havlicek[4], Thomas Kolbe[5,6], Thomas Rülicke[7], Gottfried Brem[3,8], Jo Poulton[9], Nick S Jones[1]*

[1]Department of Mathematics, Imperial College London, London, United Kingdom; [2]Biotechnology in Animal Production, Department for Agrobiotechnology, IFA Tulln, IFA Tulln, Tulln, Austria; [3]Institute of Animal Breeding and Genetics, University of Veterinary Medicine Vienna, Vienna, Austria; [4]Reproduction Centre Wieselburg, Department for Biomedical Sciences, University of Veterinary Medicine, Vienna, Austria; [5]Biomodels Austria, University of Veterinary Medicine Vienna, Vienna, Austria; [6]IFA-Tulln, University of Natural Resources and Life Sciences, Tulln, Austria; [7]Institute of Laboratory Animal Science, University of Veterinary Medicine Vienna, Vienna, Austria; [8]Biotechnology in Animal Production, Department for Agrobiotechnology, IFA Tulln, Tulln, Austria; [9]Nuffield Department of Obstetrics and Gynaecology, University of Oxford, Oxford, United Kingdom

*For correspondence: nick.
jones@imperial.ac.uk

Reviewing editor: Jodi Nunnari,
University of California, Davis,
United States

**Abstract** Dangerous damage to mitochondrial DNA (mtDNA) can be ameliorated during mammalian development through a highly debated mechanism called the mtDNA bottleneck. Uncertainty surrounding this process limits our ability to address inherited mtDNA diseases. We produce a new, physically motivated, generalisable theoretical model for mtDNA populations during development, allowing the first statistical comparison of proposed bottleneck mechanisms. Using approximate Bayesian computation and mouse data, we find most statistical support for a combination of binomial partitioning of mtDNAs at cell divisions and random mtDNA turnover, meaning that the debated exact magnitude of mtDNA copy number depletion is flexible. New experimental measurements from a wild-derived mtDNA pairing in mice confirm the theoretical predictions of this model. We analytically solve a mathematical description of this mechanism, computing probabilities of mtDNA disease onset, efficacy of clinical sampling strategies, and effects of potential dynamic interventions, thus developing a quantitative and experimentally-supported stochastic theory of the bottleneck.

## Introduction

Mitochondria are vital energy-producing organelles within eukaryotic cells, possessing genomes (mitochondrial DNA, mtDNA) that replicate, degrade and develop mutations (*Rand, 2001*; *Wallace and Chalkia, 2013*). MtDNA mutations have been implicated in numerous pathologies including fatal inherited diseases and ageing (*Lightowlers et al., 1997*; *Wallace, 1999*; *Poulton et al., 2009*; *Wallace and Chalkia, 2013*). Combatting the buildup of mtDNA mutations is of paramount importance in ensuring an organism's survival. Substantial recent medical, experimental, and media attention has focused on methods to remove (*Bacman et al., 2013*) or prevent the inheritance of (*Bredenoord et al., 2008*; *Poulton et al., 2009*; *Craven et al., 2010*; *Poulton et al., 2010*; *Burgstaller et al., 2015*) mutated mtDNA in humans.

**eLife digest** Mitochondria are structures that provide vital sources of energy in our cells. DNA contained within mitochondria encodes important mitochondrial machinery, and most human cells contain hundreds or thousands of mitochondrial DNA molecules in addition to the DNA that is stored in the nucleus. Mitochondrial DNA is inherited from mothers via the egg, and the details of this inheritance are poorly understood. This question is important because inherited mistakes in mitochondrial DNA can have detrimental consequences on health, with links to fatal diseases and many other conditions.

An unfertilised egg cell contains many copies of mitochondrial DNA molecules; some may have mutations and some may not. After fertilisation, the egg divides, the number of cells in the developing embryo increases, and the number of mitochondrial DNA molecules per cell changes. If the original egg cell contained defective mitochondrial DNA, some of these new cells end up containing more defective copies than others, leading to cell-to-cell differences in the developing embryo. This potentially allows cells with the greatest number of defective mitochondria to be eliminated. The increase in this cell-to-cell variability is called 'bottlenecking', and its mechanism remains highly debated.

Johnston et al. have now used tools from maths, statistics and new experiments to address this debate, in the light of several studies that measured the mitochondrial DNA content in developing mice. This approach allowed a new theoretical model of mitochondrial DNA during the growth of an organism to be produced, which encompasses a wide range of existing theories and allows them to be compared. This model starts from the viewpoint that the hundreds or thousands of mitochondrial DNA molecules in a cell can be thought of as a population undergoing random 'birth' and 'death', and it allows the first statistical comparison of the many proposed bottleneck mechanisms.

Johnston et al. find support for two ways that cells segregate mitochondria as they multiply, and show that the decrease in the number of mitochondrial DNA molecules during bottlenecking is flexible. This reconciles a debate amongst previous studies. These findings are confirmed using new experimental data from mice, which are genetically distinct from existing studies, illustrating the generality of the model's findings. Furthermore, an analytic mathematical description that describes in detail how bottlenecking might work is produced.

Finally, Johnston et al. provide examples using this new theoretical model to suggest therapeutic strategies for diseases caused by mitochondrial DNA mutations. Future work will need to test these suggestions, and link mathematical understanding of mitochondria with healthcare data.

One means by which organisms may ameliorate the mtDNA damage that builds up through a lifetime is through a developmental process known as *bottlenecking*. Immediately after fertilisation, a single oocyte (which may contain $>10^5$ individual mtDNAs) may have a nonzero mtDNA mutant load or *heteroplasmy* (the proportion of mutant mtDNA in the cell). As the number of cells in the developing organism increases, the intercellular population then acquires an associated *heteroplasmy variance*, that is, the variance in mutant load across the population of cells (*Figure 1A*), allowing removal of cells with high heteroplasmy and retention of cells with low heteroplasmy. Intense and sustained debate exists as to the mechanism by which this increase of heteroplasmy variance occurs. Several experimental results in mice suggest that, during development, the copy number of mtDNA per cell in the germ cell line drops dramatically to $\sim10^2$, reducing the effective population size of mitochondrial genomes (*Cree et al., 2008*; *Wai et al., 2008*). One postulated bottlenecking mechanism is that this low population size accelerates genetic drift and so increases the cell-to-cell heteroplasmy variance (*Bergstrom and Pritchard, 1998*; *Aiken et al., 2008*; *Cree et al., 2008*; *Wonnapinij et al., 2010*), which was first observed to generally increase from primordial germ cells through primary oocytes to mature oocytes (*Jenuth et al., 1996*). However, independent experimental evidence (*Wai et al., 2008*) suggests that heteroplasmy variance increases negligibly during this copy number reduction, though this interpretation has been debated (*Samuels et al., 2010*). *Wai et al. (2008)* shows heteroplasmy variance rising during folliculogenesis, after the mtDNA copy number minimum has been passed. In yet another picture, supported by conflicting experimental results (*Cao et al., 2007, 2009*), heteroplasmy variance increases with a less pronounced

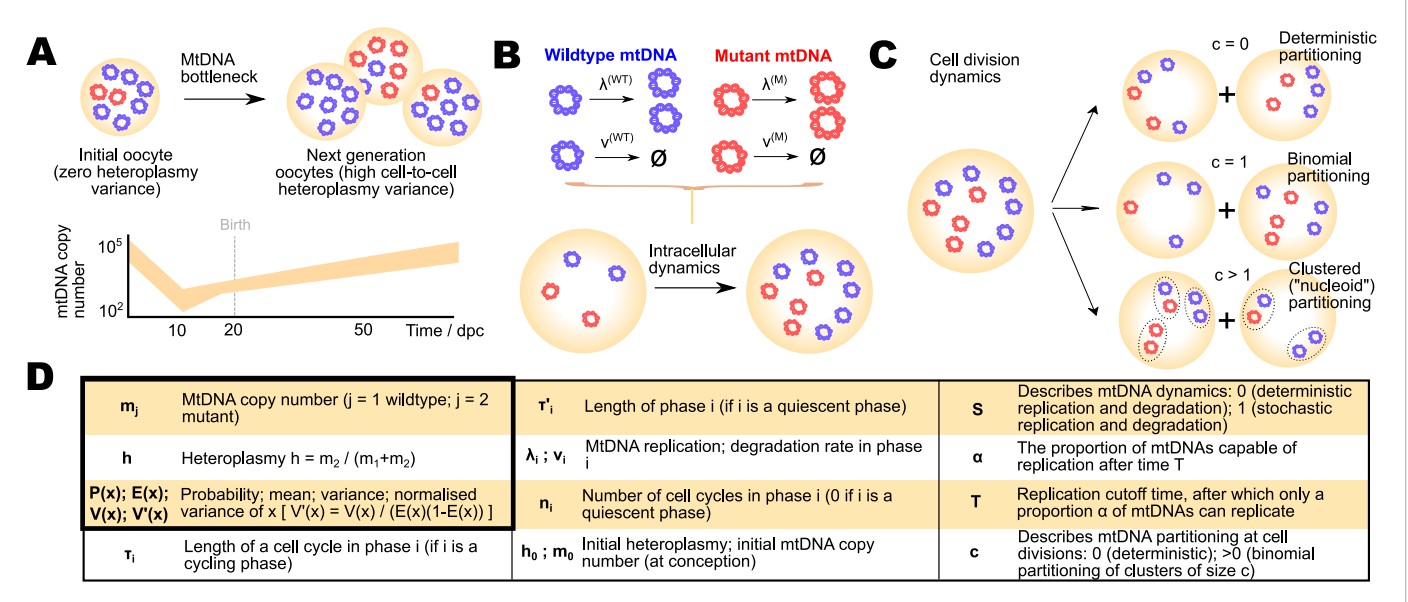

**Figure 1**. The mitochondrial bottleneck, and elements of a general model for bottlenecking mechanisms. (**A**) The mitochondrial DNA (mtDNA) bottleneck acts to produce a population of oocytes with varying heteroplasmies from a single initial oocyte with a specific heteroplasmy value. During development, mtDNA copy number per cell decreases (by a debated amount, which we address; see Main text) then recovers, suggesting a 'bottleneck' of cellular mtDNA populations. (**B**) Cellular mtDNA populations during the bottleneck are modelled as containing wildtype and mutant mtDNAs. MtDNAs can replicate and degrade within a cell cycle, with rates $\lambda$ and $\nu$ respectively. (**C**) At cell divisions, the mtDNA population is partitioned between two daughter cells either deterministically, binomially, or through the binomial partitioning of mtDNA clusters. (**D**) Symbols used to represent quantities and model parameters used in the Main text, and their biological interpretations.

decrease in mtDNA copy number (a minimum copy number $>10^3$ in mice), solely through random effects associated with partitioning at cell divisions. Clearly a consensus on this important mechanism is yet to be reached.

Important existing theoretical work on modelling the bottleneck has assumed a particular underlying mechanism (*Bergstrom and Pritchard, 1998*; *Wolff et al., 2011*) or derived statistics of mtDNA populations (*Chinnery and Samuels, 1999*; *Elson et al., 2001*; *Wonnapinij et al., 2008*, *2010*) without explicitly considering changing mtDNA population size, or the discrete nature of the mtDNA population: effects which may powerfully affect mtDNA statistics. To capture these effects it is necessary to employ a 'bottom-up' physical description of mtDNA as populations of individual, discrete elements subject to replication and degradation, as in, for example, (*Chinnery and Samuels, 1999*) and (*Capps et al., 2003*). Exploring the bottleneck also requires explicitly modelling partitioning dynamics throughout a series of cell divisions, over which population size can change dramatically. While previous simulation work (*Cree et al., 2008*; *Poovathingal et al., 2009*) has taken such a philosophy with specific model assumptions, we are not aware of such a study allowing for the wide variety of replication and partitioning dynamics proposed in the literature; we further note that replication-degradation-partitioning mtDNA models are yet to be fully described analytically. Nor is there a general quantitative framework under which different proposed bottleneck mechanisms can be statistically compared given extant data (although statistical analyses focusing on particular mechanisms and individual sets of experimental results have been used throughout the literature, for example, using a Bayesian approach under a particular bottleneck model to infer model bottleneck size [*Marchington et al., 1998*]). Combined developments in theory and inference are therefore required to make progress on this important question.

We remedy this situation by constructing a general model (features and parameters described in *Figure 1*) for the population dynamics of the bottleneck, able to describe the range of proposed mechanisms existing in the literature. Using experimental data on mtDNA statistics through development (*Jenuth et al., 1996*; *Cao et al., 2007*; *Cree et al., 2008*; *Wai et al., 2008*), we use approximate Bayesian computation (*Beaumont et al., 2002*; *Toni et al., 2009*; *Sunnåker et al., 2013*;

*Johnston, 2014*) to rigorously explore the statistical support for each mechanism, showing that random mtDNA turnover coupled with binomial partitioning of mtDNAs at cell divisions is highly likely, and that the debated magnitude of mtDNA copy number reduction is somewhat flexible. Subsequently, we confirm the predictions of this model by performing new experimental measurements of heteroplasmy statistics in mice with an mtDNA admixture, including a wild-derived haplotype, that is genetically distinct from previous studies. We then analytically solve the equations describing mtDNA population dynamics under this mechanism and show that these results allow us to investigate potential interventions to modulate the bottleneck (suggesting that upregulation of mtDNA degradation may increase the power of the bottleneck to avoid inherited disease; we discuss potential strategies for such an intervention) and yield quantitative results for clinical questions including the timescales and probabilities of disease onset, and the efficacy of strategies to sample heteroplasmy in clinical planning.

## Results

### A general mathematical model encompassing proposed bottlenecking mechanisms

We will consider three different classes of proposed generating mechanisms for the mtDNA bottleneck: those proposed in *Cao et al. (2007)*; *Cree et al. (2008)* and *Wai et al. (2008)*. We will refer to these mechanisms by their leading author name. The Cree mechanism involves random replication and degradation of mtDNAs throughout development, and binomial partitioning of mtDNAs at cell divisions. The Cao mechanism involves partitioning of clusters of mtDNA at each cell division, thus providing strong stochastic effects associated with each division. We consider a general set of dynamics through which this cluster inheritance may be manifest, including the possibility of heteroplasmic 'nucleoids' of constant internal structure (*Jacobs et al., 2000*), sets of molecules or nucleoids within an organelle, homoplasmic clusters, and different possible cluster sizes (see Appendix 1). The Wai mechanism involves the replication of a subset of mtDNAs during folliculogenesis. We note that this latter mechanism can be manifest in several ways: (a) through slow random replication of mtDNAs (so that, in any given time window, only a subset of mtDNAs will be actively replicating) or (b) through the restriction of replication to a specific subset of mtDNAs at some point during development. We will refer to these different manifestations as Wai (a) and Wai (b) respectively. The Wai (a) mechanism and the Cree model can both be addressed in the same mechanistic framework (with potentially different parameterisations): if the rate of random replication in the Cree model is sufficiently low during folliculogenesis, only a subset of mtDNAs will be actively replicating at any given time during this period, thus recapitulating the Wai (a) mechanism (see Appendix 1). We will henceforth combine discussion of the Wai (a) and Cree mechanisms into what we term the birth-death-partition (BDP) mechanism.

We seek a physically motivated mathematical model for the bottleneck that is capable of reproducing each of these mechanisms. Our general model for the bottleneck (detailed description in 'Materials and methods') involves a 'bottom-up' representation of mtDNAs as individual intracellular elements capable of replication and degradation (*Figure 1B*) with rates $\lambda$ and $\nu$ respectively. A parameter $S$ determines whether these processes are deterministic (specific rates of proliferation) or stochastic (replication and degradation of each mtDNA is a random event). These rates of replication and degradation of mtDNA are likely strongly linked to mitochondrial dynamics within cells, through the action of mitochondrial quality control (*Twig et al., 2008*; *Hill et al., 2012*) modulated by mitochondrial fission and fusion (*Detmer and Chan, 2007*; *Youle and van der Bliek, 2012*; *Hoitzing et al., 2015*), which can act to recycle weakly-perfoming mitochondria (*Mouli et al., 2009*; *Twig and Shirihai, 2011*). This quality control can be represented through the degradation rates assigned to each mtDNA species, which may differ (for selective quality control) or be identical (for non-selective turnover).

The proportion of mtDNAs capable of replication is controlled by a parameter $\alpha$ in our model, dictating the proportion of mtDNAs that may replicate after a cutoff time $T$. Thus, if $\alpha = 1$, all mtDNAs may replicate; if $\alpha < 1$, replication of a subset proportion $\alpha$ of mtDNAs is enforced at this cutoff time. At cell divisions, mtDNAs may be partitioned either deterministically, binomially, or in clusters according to a parameter $c$ (*Figure 1C*).

The copy number of mtDNA per cell is observed to vary dramatically during development, with dynamic phases of copy number depletion and different rates of subsequent recovery observed. Additionally, cell divisions occur in the germline at different rates during development, with cells becoming largely quiescent after primary oocytes develop. To explicitly model these different dynamic regimes, and the behaviour of mtDNA copy number during each, we include six different dynamic phases throughout development, each with different rates of replication and degradation (labelled with subscript $i$ labelling the dynamic phase: hence $\lambda_1, \nu_1,\ldots,\lambda_6, \nu_6$), and allowing for different rates of cell division or quiescence. This protocol enables us to explicitly model effects of changing population size throughout development rather than assuming dependence on a single, coarse-grained effective population size; and to include the effects of specific and varying cell doubling times. A summary of symbols used in our model and throughout this article is presented in *Figure 1D*.

Our model, with suitable parameterisation, can thus mirror the dynamics of the Cree and Wai (a) mechanisms (stochastic dynamics and binomial partitioning, which we refer to as the BDP mechanism); the Cao mechanism (clustered partitioning); and Wai (b) mechanism (deterministic dynamics, restricted subset of replicating mtDNAs). The Cao mechanism, partitioning of clusters of mtDNA molecules, represents the expected case if mtDNA is partitioned in colocalised 'nucleoids' within each organelle (or in other sub-organellar groupings). The size of mtDNA nucleoids is debated in the literature (*Bogenhagen, 2012*; *Kukat and Larsson, 2013*; *Wallace and Chalkia, 2013*) (although recent evidence from high-resolution microscopy suggests that nucleoid size is generally <2 (*Jakobs and Wurm, 2014*), consonant with recent evidence that individual nucleoids may be homoplasmic [*Poe et al., 2010*]); our model allows for inheritance of homoplasmic or heteroplasmic nucleoids of arbitrary characteristic size $c$, thus allowing for a range of sub-organellar mtDNA structure. We discuss the impact of mixed or fixed nucleoid content in Appendix 1.

## A BDP model of mtDNA dynamics has most statistical support given experimental measurements

We take data on mtDNA copy number in germ line cells in mice from three experimental studies (*Cao et al., 2007*; *Cree et al., 2008*; *Wai et al., 2008*). We also use data from two experimental studies on heteroplasmy variance in the mouse germ line during development (*Jenuth et al., 1996*; *Wai et al., 2008*). These heteroplasmy variance studies employ intracellular combinations of the same pairing of mtDNA haplotypes (NZB and BALB/c), modelling two different mtDNA types within a cellular population. These data, by convention (*Samuels et al., 2010*), are normalised by heteroplasmy level $h$, giving

$$\mathbb{V}'(h) = \frac{\mathbb{V}(h)}{\mathbb{E}(h)(1 - \mathbb{E}(h))}, \tag{1}$$

where normalised variance $\mathbb{V}'(h)$ is a quantity that will be often used subsequently. This normalised variance controls for the effect of different or changing mean heteroplasmy, and thus allows a comparison of heteroplasmy variance among samples with different mean heteroplasmies and subject to heteroplasmy change with time. We use a time of 100 dpc to correspond to mature oocytes (see 'Materials and methods'). We take data on cell doubling times from a classical study (*Lawson and Hage, 1994*) (see 'Materials and methods'). A possible summary of these data (although they provoke ongoing debate; see 'Discussion') is that, as shown in *Figure 2A*, the existing data on normalised heteroplasmy variance shows initially low variance until ~7.5 dpc (days post conception, which we use as a unit of time throughout), rising to intermediate values between 7.5 and 21 dpc, gradually rising further subsequently to become large in the mature oocytes of the next generation. In *Figure 2A*, and throughout this article, experimentally measured data will be depicted as circular or polygonal points, and inferred theoretical behaviour will be depicted as lines or shaded regions.

*Figure 2A* shows mtDNA population dynamic trajectories resulting from optimised parameter-isations of each of the mechanisms we consider (see 'Materials and methods'). In *Figure 2B* we show posterior probabilities on each of these mechanisms. These posterior probabilities give the inferred statistical support for each mechanism, derived from model selection performed with approximate Bayesian computation (ABC) (*Beaumont et al., 2002*; *Toni et al., 2009*; *Sunnåker et al., 2013*; *Johnston, 2014*) using uniform priors. ABC involves choosing a threshold value dictating how close a fit to experimental data is required to accept a particular model parameterisation as reasonable. In

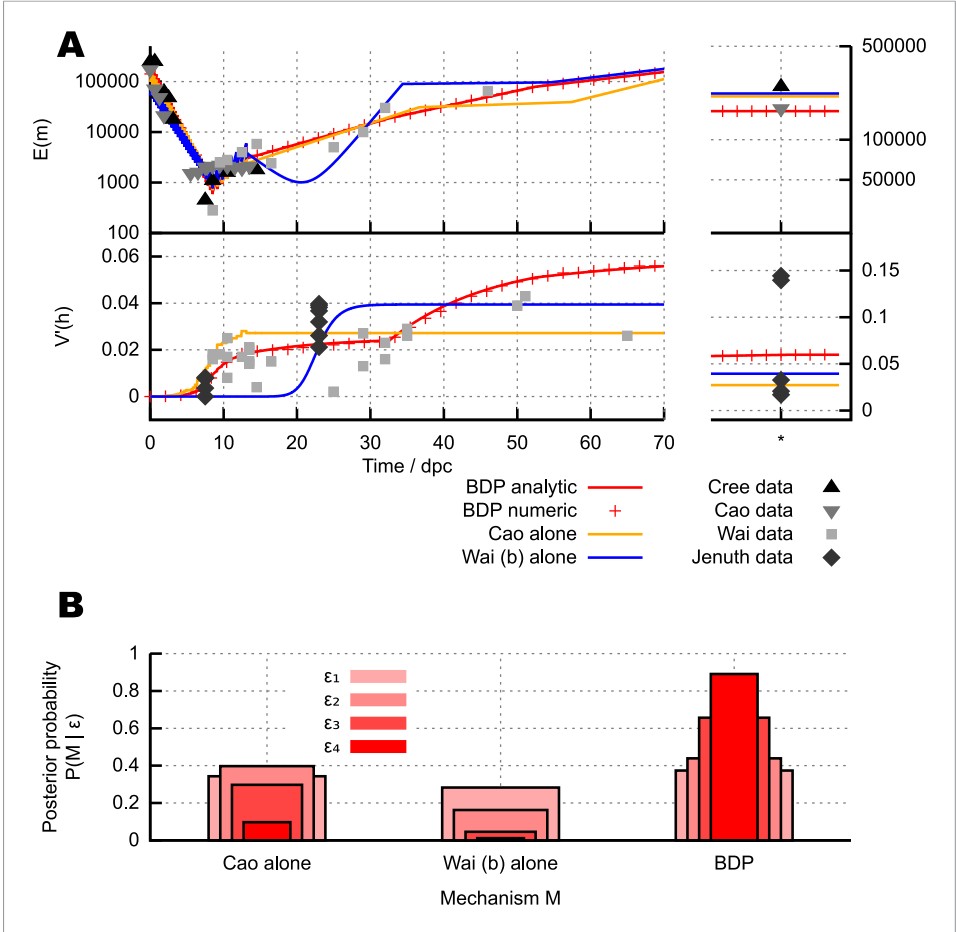

**Figure 2**. Different mechanisms for the mtDNA bottleneck. (**A**) Trajectories of mean copy number $\mathbb{E}(m)$ and normalised heteroplasmy variance $\mathbb{V}(h)$ arising from the models described in the text, optimised with respect to data from experimental studies. Birth-death-partition (BDP) denotes the BDP model, encompassing Cree and Wai (**A**) mechanisms. Left plots show trajectories during development; right plots show behaviour in mature oocytes in the next generation. * denotes measurements in mature oocytes, modelled as 100 dpc (see 'Materials and methods'). (**B**) Statistical support for different mechanisms from approximate Bayesian computation (ABC) model selection with thresholds $\epsilon_{1,2,3,4} = 75, 60, 50, 45$. As the threshold decreases, forcing a stricter agreement with experiment (thinner, darker columns), support converges on the BDP model.

our case, this goodness-of-fit is computed using a comparison of squared residuals associated with the trajectories of mean mtDNA copy number and normalised heteroplasmy variance (see 'Materials and methods' and Appendix 1). Each of the experimental measurements corresponds to a sample variance, derived from a finite number of samples of an underlying distribution of heteroplasmies, and therefore has an associated uncertainty and sampling error (*Wonnapinij et al., 2010*). The reasonably small sample sizes used in these sample variance measurements are likely to underestimate the underlying heteroplasmy variance (the target of our inference). Our ABC approach naturally addresses these uncertainties by using summary statistics derived from sampling a set of stochastic incarnations of a given model, where the size of this set is equal to the number of measurements contributing to the experimentally-determined statistic (see 'Materials and methods'). *Figure 2B* clearly shows that as the ABC threshold is decreased, requiring closer agreement between the distributions of simulated and experimental data, the posterior probability of the BDP model increases, to dramatically exceed those of the other models. This increase indicates that the BDP model is the most statistically supported, and capable of providing the best explanation of experimental data (which can be intuitively seen from the trajectories in *Figure 2A*). We note that ABC model selection automatically

takes model complexity into account, and conclude that the BDP mechanism is the best supported proposed mechanism for the bottleneck. Briefly, this result arises because the BDP model produces increasing variance both due to early cell division stochasticity *and* later random turnover. By contrast, the Cao model alone only increases variance in early development when cell divisions are occurring. Qualitatively, this behaviour through time holds regardless of cluster (nucleoid) size and regardless of whether clusters are heteroplasmic or homoplasmic (allowing heteroplasmic clusters decreases the magnitude of heteroplasmy variance but not its behaviour through time, see Appendix 1). The Wai (b) model alone similarly only increases variance at a single time point (later, during folliculogenesis).

In *Wai et al. (2008)*, visualisations of cells after BrU incorporation show that a subset of mitochondria retain BrU labelling, which the authors suggest indicates that a subset of mtDNAs are replicating. In Appendix 1, we show that the BDP model also results in the observation of only a subset of replicating mtDNAs over the timeframe corresponding to these experimental results. These observations thus correspond to results expected from the random turnover from the BDP model. We also note the mathematical observation that the Wai (b) mechanism requires the replication of <1% of mtDNAs during folliculogenesis to yield reasonable heteroplasmy variance increases (*Figure 2A* shows the optimal case with $\alpha = 0.006$; optimal fits to data generally show $0.005 < \alpha < 0.01$), and the proportions of loci visible in *Wai et al. (2008)* are substantially higher than this required 1% value.

We show in Appendix 1 that the heteroplasmy statistics corresponding to binomial partitioning also describe the case where the elements of inheritance are heteroplasmic clusters, where the mtDNA content of each cluster is randomly sampled from the population of the cell (either once, as an initial step, or repeatedly at each division). This similarity holds broadly, regardless of whether the internal structure of clusters is constant across cell divisions or allowed to mix between divisions. The BDP model, in addition to describing the partitioning of individual mtDNAs, also thus represents the statistics of mtDNA populations in which heteroplasmic nucleoids are inherited (*Jacobs et al., 2000*), or individual organelles containing a mixed set of mtDNAs or nucleoids are inherited, regardless of the size of these nucleoids (see 'Discussion').

## Parameterisation and interpretation of the BDP model

Having used ABC model selection to identify the BDP model as the most statistically supported, we can also use ABC to infer the values of the governing parameters of this model given experimental data. *Figure 3A,B* shows the trajectories of mean copy number and mean heteroplasmy variance resulting from model parameterisations identified through this process. *Figure 3C* shows the inferred behaviour of mtDNA degradation rate $\nu$ in the model, a proxy for mtDNA turnover (as the copy number is constrained). Turnover is generally low during cell divisions, allowing heteroplasmy variance to increase due to stochastic partitioning. Turnover then increases later in germ line development, resulting in a gradual increase of heteroplasmy variance after birth until the mature oocytes form in the next generation.

*Figure 3D* shows posterior distributions on the copy number minimum and total turnover (see 'Materials and methods') resulting from this process; posteriors on all other parameters are shown in Appendix 1. Substantial flexibility exists in the magnitude of the copy number minimum, illustrating that observed heteroplasmy variance can result from a range of bottleneck sizes from ~200 to >$10^3$; going some way towards reconciling the conflict between *Cao et al. (2007)* and *Cao et al. (2009)* and *Cree et al. (2008)* and *Wai et al. (2008)*. The total amount of mtDNA turnover (presented as $\sigma = \sum_{i=3}^{6} \nu_i \tau'_i$, the product of turnover rate and the time for which this rate applies, summed over quiescent dynamic phases; for example, a turnover rate of 0.1 hr$^{-1}$ for 30 days yields $\sigma = 0.1 \times 24 \times 30 = 72$) is constrained more than the specific trajectory of mtDNA turnover rates, showing that a variety of time behaviours of turnover are capable of producing the observed heteroplasmy behaviour.

## Experimental verification of the BDP model

The bottleneck mechanism identified through our analysis has several characteristic features which facilitate experimental verification. Key among these are the prediction that heteroplasmy variance acquires an intermediate (nonzero, but not maximal) value as a result of the copy number bottleneck, then continues to increase due to mtDNA turnover in later development. Our theory also produces quantitative predictions regarding the structure of heteroplasmy distributions at arbitrary times.

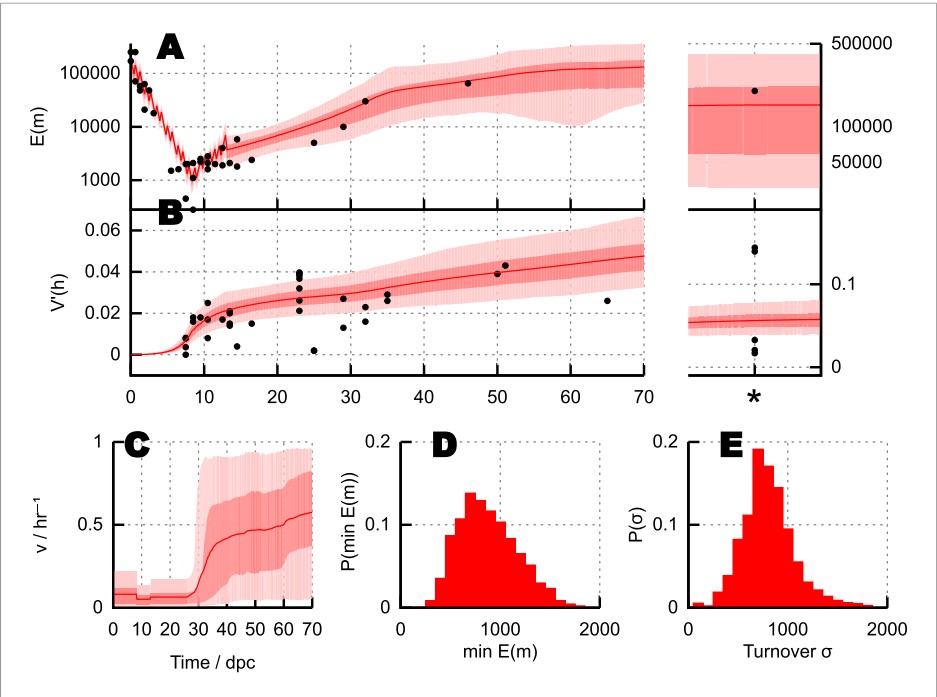

**Figure 3**. Parameterisation of the BDP model and inferred details of bottleneck mechanism. Trajectories of (**A**) mean copy number $\mathbb{E}(m)$ and (**B**) normalised heteroplasmy variance $\mathbb{V}'(h)$ resulting from BDP model parameterisations sampled using ABC with a threshold $\epsilon = 40$. * denotes measurements in mature oocytes, modelled as 100 dpc (see 'Materials and methods'). *Note: the range in (**B**) does not correspond to a credibility interval on individual measurements, but rather on an expected underlying (population) variance, from which individual variance measurements are sampled.* We thus expect to see, for example, several measurements lower than this range due to sampling limitations (see text). (**C**) Posterior distributions on mtDNA turnover $\nu$ with time. (**D**) Posterior distribution on min $\mathbb{E}(m)$, the minimum mtDNA copy number reached during development. (**E**) Posterior distribution on $\sigma = \sum_{i=3}^{6} \tau'_i \nu_i$, a measure of the total amount of mtDNA turnover.

The existing data that we used to perform inference and model selection display a degree of internal heterogeneity, coming from several different experimental groups. Furthermore, these data represent statistics resulting from a single pairing of mtDNA types, and it is thus arguable how conclusions drawn from them may represent the more genetically diverse reality of biology. *Burgstaller et al. (2014)* recently addressed this issue of a limited number of mtDNA pairings by producing novel mouse models involving mixtures of standard and several new, unexplored, wild-derived haplotypes which capture a range of genetic diversity. To test the applicability and generality of our predictions, we have perfomed new experimental measurements of germline heteroplasmy variance in these model animals under a consistent experimental protocol (see 'Materials and methods'). We use the 'HB' mouse line from *Burgstaller et al. (2014)* pairing a wild-derived mtDNA haplotype (labelled 'HB' after its source in Hohenberg, Germany) with C57BL/6N; we refer to this model as 'HB'.

Heteroplasmy measurements were taken in oocytes sampled from mice at ages 24–61 dpc (see 'Materials and methods' and Appendix 1; raw data in *Figure 4—source data 1*). The statistics of these measurements yielded $\mathbb{E}(h)$, $\mathbb{V}(h)$ and $\mathbb{V}'(h)$ as previously. This age range was chosen to address the regions with most power to discriminate between the competing models; the existing $\mathbb{V}'(h)$ data is most heterogeneous around 20–30 dpc and the later datapoints allow us to detect developmental heteroplasmy behaviour after the copy number minimum. *Figure 4A* shows these $\mathbb{V}'(h)$ measurements. The qualitative behaviour predicted by the BDP mechanism is clearly visible: variance around birth (after the copy number bottleneck) is low but non-zero, subsequently increasing with time. The ability of the BDP model to account for the magnitudes and time behaviour of heteroplasmy variance

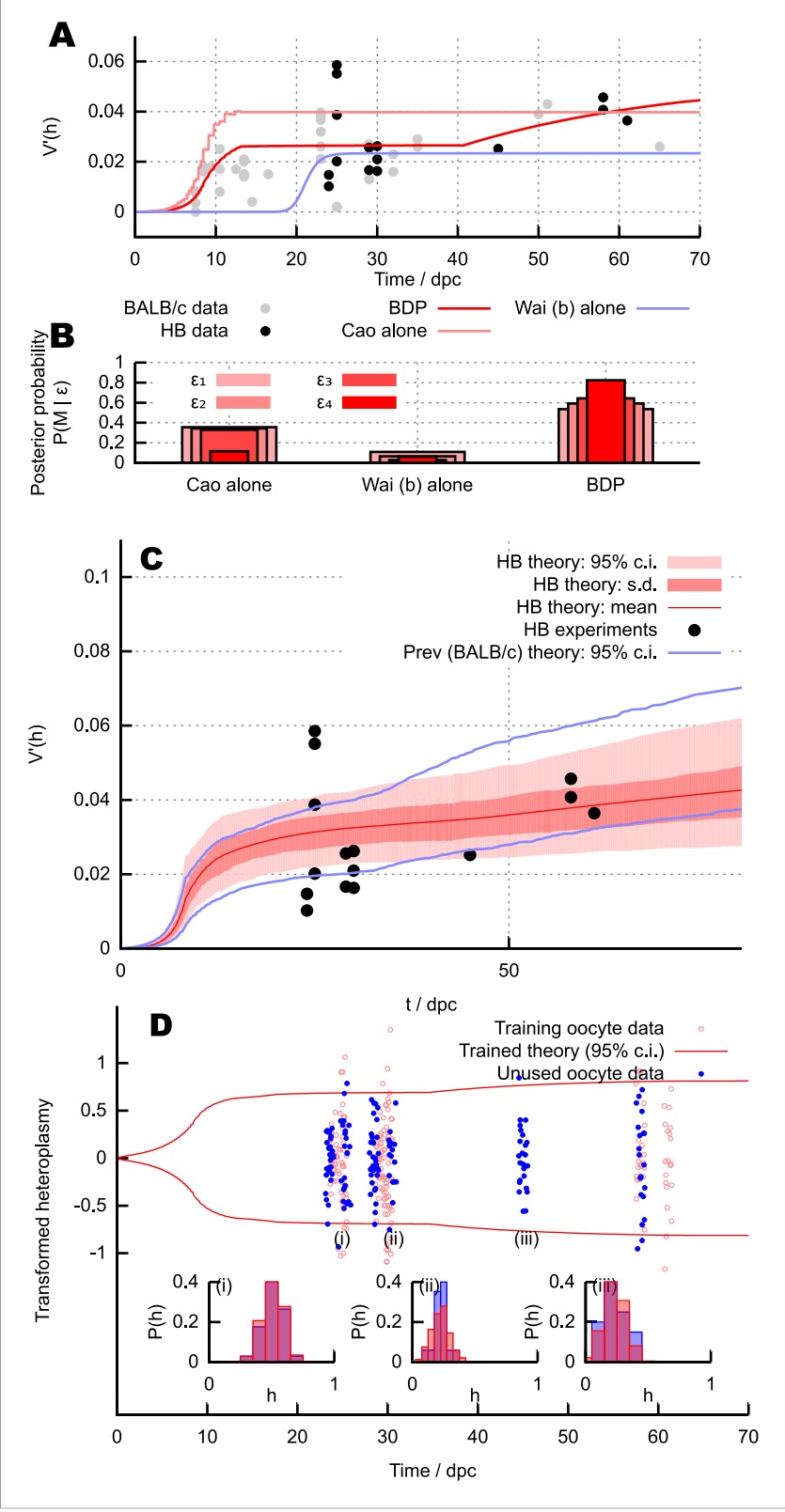

**Figure 4.** Predictions and experimental verification of the BDP model. (**A**) New $\mathbb{V}'(h)$ measurements from the HB mouse system, with optimised fits for the BDP, Wai (b) and Cao models. (**B**) Posterior probabilities of each model given this data under decreasing ABC threshold: $\epsilon = \{50, 40, 30, 25\}$. (**C**) All $\mathbb{V}'(h)$ measurements from the HB model (points) with inferred $\mathbb{V}'(h)$ behaviour from ABC applied to the BDP model (red curves). *As in* **Figure 3**, *this range*

*Figure 4. continued on next page*

*Figure 4. Continued*

does not correspond to a credibility interval on individual measurements, but rather on an expected underlying (population) variance, from which individual variance measurements are sampled. The inferred behaviour strongly overlaps with the inferred behaviour for the BALB/c system (blue curves), suggesting that the BDP model applies to a genetically diverse range of systems. (**D**) Heteroplasmy distributions. The transformation $h' = -\ln\left|(h^{-1} - 1)\mathbb{E}(h)/(1 - \mathbb{E}(h))\right|$ (*Burgstaller et al., 2014*) is used to compare distributions with different mean heteroplasmy. Red jitter points are samples from sets used to parameterise the BDP model; red curves show the 95% range on transformed heteroplasmy with time inferred from these samples. Blue jitter points are samples withheld independent from this parameterisation; their distribuuions fall within the independently inferred range. Insets show, in untransformed space, distributions of the withheld heteroplasmy measurements (blue) compared to parameterised predictions (red); no withheld datasets show significant support against the predicted distribution (Anderson-Darling test, *p* < 0.05).

The following source data is available for figure 4:

**Source data 1**. Individual heteroplasmy measurements in the HB mouse model contributing to the new heteroplasmy variance data used to test our theory.

more satisfactorily than the alternative models is shown by the model fits in *Figure 4A*. We explored these new data quantitatively through the same model selection approach used for the existing data. As shown in *Figure 4B*, the BDP mechanism again experiences by far the strongest statistical support in this genetically different system.

*Figure 4C* shows the result of our parameteric inference approach using these $\mathbb{V}'(h)$ measurements coupled with the $\mathbb{E}(m)$ measurements used previously (employing our assumption that modulation of copy number by heteroplasmy in this non-pathological haplotype is small). Strikingly, the quantitative behaviour of $\mathbb{V}'(h)$ with time inferred from the HB model (red) matches the previous behaviour inferred from the NZB/BALB/c system (blue) very well, suggesting that our theory is applicable across a range of genetically distinct pairings. We note that the shaded region in *Figure 4C* corresponds to credibility intervals around the *mean* behaviour of $\mathbb{V}'(h)$, and the fact that individual $\mathbb{V}'(h)$ datapoints (subject to fluctuations and sampling effects) do not all lie within these intervals is not a signal of poor model choice. An analogous situation is the observation of a scatter of datapoints outside the range of the standard error on the mean (s.e.m.), which does not imply a mistake in the s.e.m. estimate. The difference between the trace in *Figure 4A* and the mean curve in *Figure 4C* arises because *Figure 4A* shows the behaviour of the model under a single, optimised parameterisation, whereas *Figure 4C* shows the distribution of model behaviours over the posterior distributions on parameters: the mean $\mathbb{V}'(h)$ trace of this distribution is comparable but not equivalent to that from the single best-fit parameterisation.

To confirm more detailed predictions of our model, we also examined the specific distributions of heteroplasmy in our new measurements. Given a mean heteroplasmy and an organismal age, the parameterised BDP model predicts the structure of the heteroplasmy distribution (see 'Materials and methods' and next section). We parameterised the model using $\mathbb{V}'(h)$ values from a subset of half of the new measurements (chosen by omitting every other sampled set when ordered by time). *Figure 4D* shows a comparison of measured heteroplasmy distributions with a 95% bound from the parameterised BDP model. We then tested the predictions of the parameterised model against the other half of new measurements. 8 of the test measurements (2.4%) fell outside the inferred 95% bound from the training dataset, illustrating a good agreement with distributional predictions. The Anderson-Darling test was used to compare the distribution of heteroplasmy in sampled oocytes with distributions predicted by our theory (given age and mean heteroplasmy); no set of samples showed significant (*p* < 0.05) departures from the hypothesis that the two distributions were identical. Some example distributions are presented in *Figure 4D* (i), (ii), (iii).

## The BDP model is analytically tractable

Importantly, the BDP model yields analytic solutions for the values of all genetic properties of interest, using tools from stochastic processes (detail in 'Materials and methods' and Appendix 1). These results facilitate straightforward further study and fast predictions of timescales and probabilities of interest. The full theoretical approach is detailed in Appendix 1, and equations for

the mean and variance of mtDNA populations and heteroplasmy are given in the Methods. In *Figure 2A* we illustrate that these analytic results provide an excellent match to the numeric results of stochastic simulation, a result that holds across all BDP model parameterisations. It is also straightforward to calculate the fixation probability $\mathbb{P}(m=0)$, which allows us to characterise all heteroplasmy distributions that arise from the bottlenecking process, even when highly skewed (see 'Materials and methods' and Appendix 1). We have thus obtained analytic solutions for the time behaviour of mtDNA copy number and heteroplasmy throughout the bottleneck with no assumptions of continuous population densities or fixed population size, under a physical model with the most statistical support given experimental data.

## Mitochondrial turnover, degradation, and selective pressures exert quantifiable influence on heteroplasmy variance

We can use our theory to explore the dependence of bottleneck dynamics on specific biological parameters. We first explore the effects of modulating mtDNA turnover by varying $\lambda$ and $\nu$ in concert, corresponding to an increase in mtDNA degradation balanced by a corresponding increase in mtDNA replication. This increased mtDNA turnover increases the heteroplasmy variance (see *Figure 5A*) due to the increased variability in mtDNA copy number from the underlying random processes occurring at increased rates. We find that increasing mtDNA degradation $\nu$ without increasing $\lambda$ also increases heteroplasmy variance, in addition to decreasing the overall mtDNA copy number (*Figure 5B*). Applying this unbalanced increase in mtDNA degradation without a matching change in replication has a strong effect on mtDNA dynamics as it corresponds to a universal change in the 'control' applied to the system, analogous, for example, to changing target copy numbers in manifestations of relaxed replication (*Chinnery and Samuels, 1999*). The simple model we use does not include feedback, and controls mtDNA dynamics solely through kinetic parameters. Perturbing the balance of these parameters thus strongly affects the expected behaviour of the system. As we discuss later, elucidation of the specific mechanisms by which control is manifest in mtDNA populations will require further research, but these numerical experiments attempt to represent the cases where a perturbation is naturally compensated for (matched changes, *Figure 5A*) and where it is not (unbalanced change, *Figure 5B*).

These results suggest that an artificial intervention increasing mitochondrial degradation may generally be expected to increase heteroplasmy variance during development. An increase in mtDNA degradation is expected to either directly increase heteroplasmy variance (*Figure 5B*) if mtDNA populations are weakly controlled, or to provoke a compensatory, population-maintaining increase in mtDNA replication, thus increasing mtDNA turnover, which also acts to increase variance (*Figure 5C*) if mtDNA populations are subject to feedback control. The increase in variance through either of these pathways will increase the power of cell-level selection to remove cells with high heteroplasmy and thus purify the population. For this reason, we speculate that mitochondrial degradation may represent a potential clinical target to address the inheritance of mtDNA disease (more detail in Appendix 1).

Our model also allows us to explore the effect of different mtDNA types experiencing different selective pressures, by setting $\lambda_1 \neq \lambda_2$ (mutant mtDNA experiences a proliferative advantage or disadvantage). Such a selective difference causes changes in both mean heteroplasmy and heteroplasmy variance, as shown in *Figure 5C* (e.g., if heteroplasmy decreases towards zero, heteroplasmy variance will also decrease, as the wildtype is increasingly likely to become fixed). We do not focus further on selection in this study, noting that selective pressures are likely to be specific to a given pair or set of mtDNA types and are not generally characterised well enough to perform satisfactory inference. However, we do note that our theory gives a straightforward prediction for the functional form of mean heteroplasmy when nonzero selection is present, a sigmoid with slope set by the fitness difference (see 'Materials and methods').

## Probabilities of exceeding threshold heteroplasmy values

A key feature of mtDNA diseases is that pathological symptoms usually manifest when heteroplasmy in a tissue exceeds a certain threshold value, with few or no symptoms manifested below this threshold (*Rossignol et al., 2003*). The probability and timescale with which cellular heteroplasmy may be expected to exceed a given value is thus a quantity of key interest in clinical planning of mtDNA disease strategies.

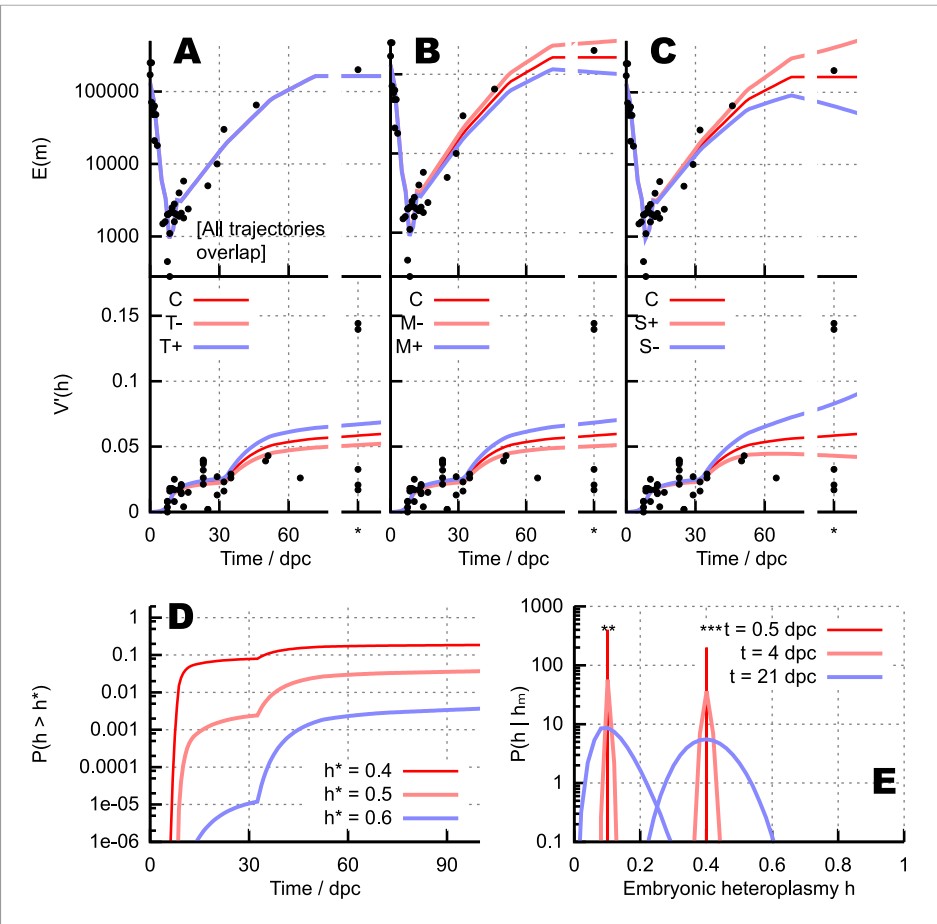

**Figure 5**. Quantitative influences and clinical results from our bottlenecking model. (**A–C**) Trajectories of copy number $\mathbb{E}(m)$ and normalised heteroplasmy variance $\mathbb{V}'(h)$ resulting from perturbing different physical parameters. Trajectory C labels the 'control' trajectory resulting from a fixed parameterisation; black dots show experimental data; * denotes measurements from primary oocytes, modelled at 100 dpc. (**A**) Increasing ($T^+$) and decreasing ($T^-$) mtDNA turnover (both mtDNA replication and degradation) by 20%. (**B**) Increasing ($M^+$) and decreasing ($M^-$) mtDNA degradation throughout development by a constant value ($2 \times 10^{-4}$, in units of day$^{-1}$), while keeping replication constant. (**C**) Applying a positive ($S^+$) and negative ($S^-$) selective pressure to mutant mtDNA by $5 \times 10^{-6}$ day$^{-1}$. (**D**) Probability of crossing different heteroplasmy thresholds $h^*$ with time, starting with initial heteroplasmy $h_0 = 0.3$. (**E**) Probability distributions over embryonic heteroplasmy $h$ given a measurement $h_m$ from preimplantation sampling (** $h_m = 0.1$; *** $h_m = 0.4$) at different times.

In our model, the probability, as a function of time, of a cell containing $m_1$ wildtype and $m_2$ mutant mtDNAs can be straightforwardly derived. The resultant analytic expression involves a hypergeometric function, also an important mathematical element in expressions describing mtDNA statistics based on classical population genetics (*Kimura, 1955*; *Wonnapinij et al., 2008*). The probability of obtaining a given heteroplasmy can therefore be computed as a sum over all copy number states that correspond to that heteroplasmy. However, as hypergeometric functions are comparatively unintuitive and computationally expensive, we here employ an approximation to the distribution of heteroplasmy based upon the above moments that are straightforwardly calculable from our model. This approximation involves fixation probabilities for each mtDNA type and a truncated Normal distribution for intermediate heteroplasmies (see 'Materials and methods'). In Appendix 1 we show that this approximation corresponds well to the exact distributions calculated using the hypergeometric function. We underline that exact heteroplasmy distributions are straightfoward to compute using our approach: we use the truncated Normal approximation as it represents the exact distribution well, is more intuitively interpretable, and is computationally very inexpensive.

Using this approach, the probability with time of a cell exceeding a threshold heteroplasmy $h^*$ can be straightforwardly computed for any initial heteroplasmy, allowing rigorous quantitative elucidation of this important clinical quantity (see 'Materials and methods'). *Figure 5D* illustrates this computation by showing the analytic probability with which thresholds $h^* = 0.4, 0.5, 0.6$ are exceeded at a time $t$, given the example initial heteroplasmy $h = 0.3$. These results serve as a simple example of the power of our modelling approach: any other specific case can readily be addressed. Our theory thus allows general quantitative calculation of the probability (and timescale) that any given heteroplasmy threshold will be exceeded, given knowledge of the initial (or early) heteroplasmy.

## Developmental sampling of embryonic heteroplasmy

We next turn to the question of estimating heteroplasmy levels in a developed organism by sampling cells during development. This principle, clinically termed preimplantation genetic diagnosis (*Steffann et al., 2006*; *Poulton et al., 2009*), assists in clinical planning by allowing inference of the specific heteroplasmic nature of the embryo itself rather than a population average of an affected mother's oocytes (*Treff et al., 2012*). However, the complicated and stochastic nature of the bottleneck makes this inference a challenging problem.

Given a heteroplasmy measurement from sampling $h_m$, accurate preimplantation diagnosis is contingent on knowledge of the distribution $\mathbb{P}(h|h_m)$, that is, the probability that the embryonic heteroplasmy is $h$ given that a measurement $h_m$ has been made. We can use our modelling framework and Bayes' theorem (see 'Materials and methods') to obtain a formula for this conditional probability, allowing a rigorous probability to be assigned to inferences from preimplantation sampling. Here, as above, we employ the truncated Normal approximation for the heteroplasmy distribution, noting that the exact treatment using hypergeometric functions is straightforward but more computationally expensive. *Figure 5E* illustrates this process by showing the probability distributions on embryonic heteroplasmy when measurements $h_m = 0.1$ or $0.4$ have been taken at different times during development. The increasing heteroplasmy variance through development means that substantially greater uncertainty is associated with heteroplasmy values inferred using measurements taken at later times. In conclusion, although care must be taken in applying this reasoning to cell types in which, for example, mitochondrial and cell turnover rates differ from those assumed here, or differentiation leads to tissue-specific selective factors acting on the mtDNA population, this formalism provides a general means of rigorously inferring embryonic heteroplasmy through genetic diagnosis sampling.

## Discussion

We have used a general stochastic model and approximate Bayesian computation with the available experimental data on developmental mtDNA dynamics to show that the bottleneck is most likely manifest through stochastic mtDNA dynamics and partitioning, with increased random turnover later during development, a mechanism which we can describe exactly and analytically (*Figure 6*). We emphasise that the bottom-up construction of our model from physical first principles both increases the flexibility and generality of our model, allowing different mechanisms to be compared together, and providing information on mtDNA dynamics throughout development rather than estimating an overall effect. We note that even though our model cannot represent the full microscopic truth underlying the mtDNA bottleneck, its ability to recapitulate the wide range of extant experimental measurements suggest that its study may yield useful insights into the effects of different treatments and perturbations on the bottleneck.

A key debate in the literature has focussed on the magnitude of the bottleneck. Some studies (*Aiken et al., 2008*; *Cree et al., 2008*) have observed a depletion of mtDNA copy number during the bottleneck to minima around several hundred; other studies (*Cao et al., 2007*, *2009*) have observed that mtDNA copy number remains $>10^3$. Our study shows that observed increases in heteroplasmy variance (*Jenuth et al., 1996*; *Wai et al., 2008*) can be achieved across this range of potential minimal mtDNA copy numbers, meaning that the much-debated magnitude of mtDNA copy number reduction is not the sole critical feature of the bottleneck, in agreement with arguments from *Cao et al. (2007, 2009)*; *Wai et al. (2008)*. We find that the role of stochastic mtDNA dynamics can play a key role in determining heteroplasmy variance without additional mechanistic details, in keeping with approaches proposed by *Cree et al. (2008)*. The mechanism with the most statistical support is thus consistent with aspects from all existing proposals in the literature.

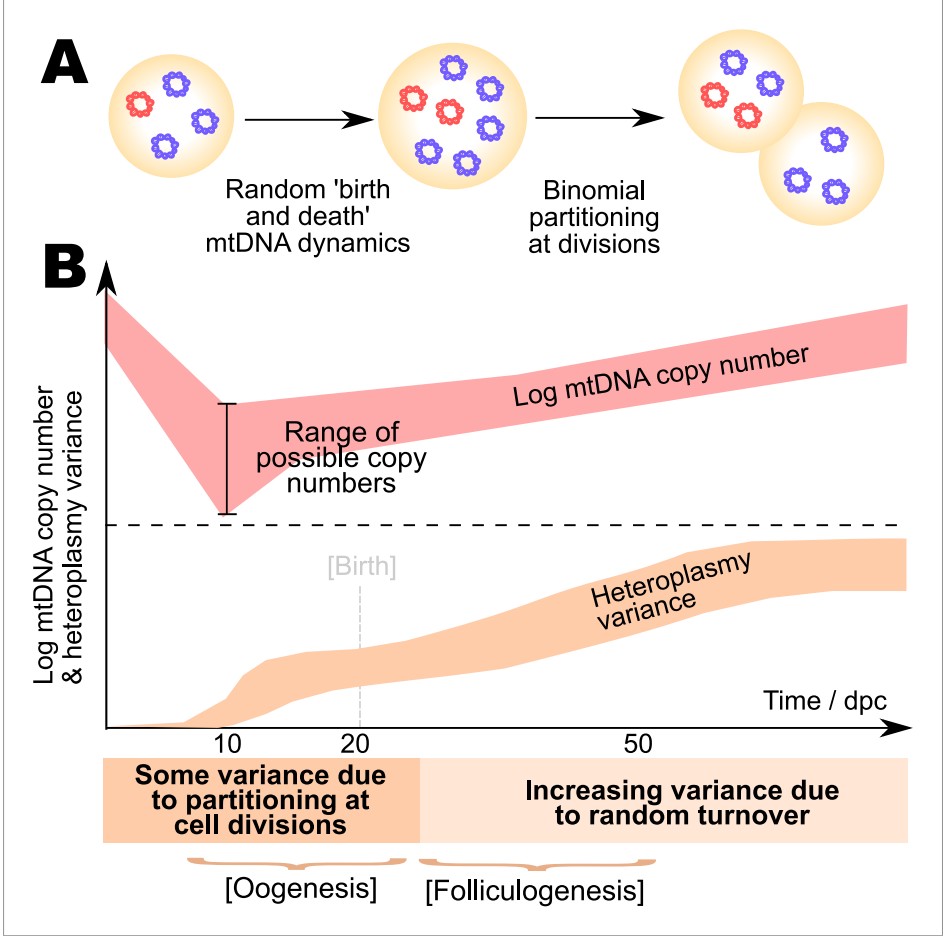

**Figure 6**. Model for the mtDNA bottleneck. A summary of our findings. (**A**) There is most statistical support for a bottlenecking mechanism whereby mtDNA dynamics is stochastic within a cell cycle, involving random replication and degradation of mtDNA, and mtDNAs are binomially partitioned at cell divisions. (**B**) This mechanism results in heteroplasmy variance increasing both due to stochastic partitioning at divisions and due to random turnover. The absolute magnitude of the copy number bottleneck is not critical: a range of bottleneck sizes can give rise to observed dynamics. Random turnover of mtDNA increases heteroplasmy variance through folliculogenesis and germline development.

We have shown that, of the models proposed in the literature, a BDP model, proposed after *Cree et al. (2008)* and compatible with an interpretation of *Wai et al. (2008)*, is the individually most likely mechanism, and capable of producing experimentally observed heteroplasmy behaviour. We cannot, given current experimental evidence, discount hybrid mechanisms, where BDP dominates the population dynamics but small contributions from other mechanisms provide perturbations to this behaviour, and propose experiments to conclusively distinguish between these cases (see Appendix 1). As the expected statistics of mtDNA populations undergoing inheritance of heteroplasmic mtDNA clusters is very similar to those undergoing binomial partitioning of mtDNAs (see Appendix 1), the inheritance of heteroplasmic nucleoids (as opposed to individual mtDNAs) is not excluded by our findings, though other recent experimental evidence suggests that this situation may be unlikely (*Poe et al., 2010*; *Jakobs and Wurm, 2014*). We contend that the most likely situation may involve the partitioning of individual organelles, containing a mixture of homoplasmic nucleoids of characteristic size <2. Notably, this case (inheritance of heteroplasmic groups, likely with fluid structure due to mixing of organellar content and mitochondrial dynamics), gives rise to statistics which our binomial model reproduces (see Appendix 1).

As mentioned in the model description, it is likely that mitochondrial dynamics (fission and fusion of mitochondria) (*Detmer and Chan, 2007*) play a role in determining natural mtDNA turnover, and

particularly mtDNA turnover in the presence of pathological mutations (*Nunnari and Suomalainen, 2012*), through the mechanism of mitochondrial quality control (*Twig et al., 2008*; *Twig and Shirihai, 2011*). Mitochondrial dynamics may also influence the elements of partitioning, through changes in the connectivity of the mitochondrial network. In our current model, these influences are coarse-grained into descriptions of the dynamic rates of mtDNA replication and degradation, and the characteristic elements that are partitioned at divisions. These physical parameters, as opposed to the more microscopic details of mitochondrial dynamics, are expected to be the key determinants of heteroplasmy statistics through development. Accounting for how these parameters, which summarize the relevant outputs of mitochondrial dynamics, connect to details of microscopic models of mitochondrial dynamics is an important future research direction to be addressed when more quantitative data is available.

The experimental data used to parameterise the first part of our study was taken from four studies in mice. Observation of similar dynamics in salmon (*Wolff et al., 2011*) points towards the bottleneck being a conserved mechanism in vertebrates. We also note that our results in mice are broadly consistent with findings from recent experiments in other organisms, suggesting that in primates and humans, heteroplasmy variance may increase at early developmental stages (*Monnot et al., 2011*; *Lee et al., 2012*), and that partitioning of mitochondria is binomial in HeLa cells (*Johnston et al., 2012*). As more studies become available on human mtDNA behaviour during development we will test our model's applicability and its clinical predictions. We note that the results of a recent study of human preimplantation sampling (*Treff et al., 2012*) found that earlier measurements provided strong predictive power of mean heteroplasmy, about which substantial variation was recorded in the offspring—both of which results are consistent with the application of our model to theoretical sampling considerations. In addition, recent observations that the m.3243A > G mutation in humans both increases mtDNA copy number during development (*Monnot et al., 2013*), and displays a less pronounced increase of heteroplasmy variance (*Monnot et al., 2011*) than other mutations, are consistent with the link between heteroplasmy variance and mtDNA copy number in our theory.

The combination of modern stochastic and statistical treatments that we have employed provides a generalisable and powerful way to recapitulate experimental data and rigorously deduce underlying biological mechanisms. We have used this combination to explore pertinent questions regarding the mtDNA bottleneck (and others have used a similar philosophy to numerically explore mtDNA point mutations [*Poovathingal et al., 2009*]): we hope to convince the reader that such methodology may be appropriate to explore other problems involving stochastic biological systems. We have used new experimental measurements to confirm our theoretical findings, illustrating the beneficial and powerful coupling of mathematical and experimental approaches to address competing hypotheses in the literature. Our detailed elucidation of the bottleneck allows us to propose further experimental methodology to address the current unknowns in our theory, including the specifics of mtDNA partitioning at cell division and the roles of selective differences between mtDNA types; importantly, we also propose a strategy to investigate our claim that our most supported model is compatible with the subset-replication picture of mtDNA dynamics. We list these experiments in full in Appendix 1. Finally, we believe that the theoretical foundation for mtDNA dynamics that we have produced allows increased quantitative rigour in the predictions and strategies involved in mtDNA disease therapies, illustrated by the above application of our theory to problems in mtDNA sampling strategies, disease onset timescales, and interventions to increase the power of the bottleneck.

## Materials and methods

### General model for mtDNA dynamics

Our 'bottom-up' model represents individual mtDNAs as elements which replicate and degrade either randomly or deterministically according to the model parameterisation. Consonant with experimental studies showing that it is often a single mutant genotype that dominates the non-wildtype mtDNA population of a cell (*Khrapko et al., 1999*), we consider two mtDNA types (wildtype and mutant), though our model can readily be extended to more mtDNA types. We denote the number of 'wildtype' mtDNAs in a cell as $m_1$ and the number of 'mutant' mtDNAs as $m_2$. The heteroplasmy of a cell is then $h = \dfrac{m_2}{m_1 + m_2}$, that is, the population proportion of mutant mtDNA.

## MtDNA dynamics within a cell cycle

Individual mtDNAs are capable of replication and degradation, with rates denoted $\lambda$ and $\nu$ respectively. According to a binary categorical parameter $S$, these events may be deterministic ($S = 0$; the mtDNA population replicates and degrades by a fixed amount per unit time) or Poisson processes ($S = 1$; each individual mtDNA randomly replicates and degrades with average rates $\lambda$ and $\nu$). A parameter $\alpha$ controls the proportion of mtDNAs capable of replication: $\alpha = 1$ allows all mtDNAs to replicate throughout development, $\alpha < 1$ enforces a subset proportion $\alpha$ of replicating mtDNAs a time cutoff $T$ after conception.

## MtDNA dynamics at cell divisions

A parameter $c$ (cluster size; a non-negative integer) dictates the partitioning of mtDNAs at cell divisions. When $c = 0$, partitioning is deterministic, so each daughter cell receives exactly half of its parent's mtDNA. For $c > 0$, partitioning is stochastic. When $c = 1$, partitioning is binomial: each mtDNA has a 50% chance of being inherited by either daughter cell. When $c > 1$, the parent cell's mtDNAs are grouped in clusters of size $c$ before division. Each cluster is then partitioned binomially, with a 50% chance of being inherited by either daughter cell.

## Different dynamic phases through development

The mtDNA population changes in different ways as development progresses, first decreasing, then recovering, then slowly growing. We include the possibility of different 'phases' of mtDNA dynamics in our model to capture this behaviour. Each phase $j$ has its own associated pairs of $\lambda_j, \nu_j$ parameters and may either be quiescent (involving no cell divisions) or cycling (encompassing $n_j$ cell divisions). Thus, we may have an initial cycling phase with low mtDNA replication rates, so that copy number falls for several cell divisions, then a subsequent 'recovery' cycling phase with higher replication rates so that mtDNA levels are amplified, then quiescent phases as cell lineages are identified. We allow six different phases, with the first two fixed as cycling phases with the above doubling times, and the final phase fixed to include no mtDNA replication (representing the stable, final oocyte state).

## Initial conditions

The initial conditions of our model involve an initial mtDNA copy number $m_0$ (the total number of mtDNAs in the fertilised oocyte) and an initial heteroplasmy $h_0$ (the fraction of these mtDNAs that are mutated).

## Data acquisition

We used three datasets for mtDNA copy number during mouse development: *Cao et al. (2007)*; *Cree et al. (2008)*; and *Wai et al. (2008)*. We use two datasets for heteroplasmy variance during development: *Wai et al. (2008)* and *Jenuth et al. (1996)*. By convention, we use the normalised versions of heteroplasmy variance (i.e., measured variance divided by a factor $h(1 - h)$). Where the measurements were not given explicitly in these publications, we manually analysed the appropriate figures to extract the numerical data. For these values, we used data from correspondence regarding the Wai study (reply to [*Samuels et al., 2010*]), and manually normalise the Jenuth dataset. The Jenuth dataset contains measurements taken in 'mature oocytes' with no time given; we assume a time of 100 dpc for these measurements, though this time is generalisable and does not qualitatively affect our results. All values are presented in Appendix 1. Data on cell doubling times in the mouse germ line is taken from *Lawson and Hage (1994)*, suggesting that doubling times start with an interval of every 7 hr, then after around 8.5 days post conception (dpc) increase to 16 hr, before the onset of a quiescent regime around 13.5 dpc (roughly consistent with the estimate of ~25 divisions between generations in the female mouse germ line [*Drost and Lee, 1995*]).

## Simulation, model selection, and parametric inference

We use Gillespie algorithms, also known as stochastic simulation algorithms (*Gillespie, 1977*), to explore the behaviour of our model of the bottlenecking process for a given parameterisation. For a given model parameterisation, the Gillespie algorithm is used to simulate an ensemble of $10^3$ possible realisations of the time evolution of mtDNA content, and the statistics of this ensemble are recorded. The experimental data we use is derived from sets of measurements of different sizes; to compare simulation data with an experimental datapoint $i$ corresponding to a statistic derived from $n_i$

measurements, we sampled a random subset of $n_i$ of the $10^3$ simulated trajectories (all datapoints but one have $n \ll 10^3$), and used this subset to derive the simulated statistic for comparison to datapoint $i$ (*Johnston, 2014*).

To fit the different models to experimental data we define a distance measure, a sum-of-squares residual between the $\mathbb{E}(m)$ (in log space) and $\mathbb{V}(h)$ dynamics produced by our model and observed in the data, weighted to facilitate comparison of these different quantities (*Johnston, 2014*). We also constrain copy number to be $<5 \times 10^5$ at all points throughout development, rejecting parameterisation that disobey this criterion. Metropolis MCMC was used to identify the best-fit parameterisation according to this distance function. For statistical inference, we use approximate Bayesian computation (ABC), a statistical approach that has successfully been applied to parametric inference and model selection in dynamical systems (*Toni et al., 2009*) to infer posterior probability distributions both for individual models and the parameters of the models given experimental data. ABC samples posterior probability distributions on parameters that lead to behaviour within a certain threshold distance of the given data; these posteriors are shown to converge on the true posteriors as the threshold value decreases to zero (see Appendix 1). We employed an MCMC sampler with randomly-selected initial conditions. For further details, including priors, thresholds and step sizes used in ABC, see Appendix 1. Minimum copy number was recorded directly from the resulting trajectories; our measure of total turnover $\sigma$ is defined as $\sigma = \sum_{i=3}^{6} \tau'_i \nu_i$, the sum over quiescent dynamic phases of the product of degradation rate and phase length.

## Creation of heteroplasmic mice

Heteroplasmic mice were obtained from a heteroplasmic mouse line (HB) we created previously by ooplasmic transfer (*Burgstaller et al., 2014*). This mouse line contains the nuclear DNA of the C57BL/6N mouse, and mtDNAs both of C57BL/6N and a wild-derived house mouse. Both mtDNA variants belong to the same subspecies, *Mus musculus domesticus*. For details on sequence divergence (see *Burgstaller et al., 2014*).

## Isolation and lysis of oocytes

Mice were sacrificed at the indicated ages by cervical dislocation. Ovaries were extracted and immediately placed in cryo-buffer containing 50% PBS, 25% ethylene glycol and 25% DMSO (Sigma–Aldrich, Austria) and stored at −80°C. For oocyte extraction, ovaries were placed into a drop of cryo-buffer and disrupted using scalpel and forceps. Oocytes were collected and remaining cumulus cells were removed mechanically by repeated careful suction through glass capillaries. Prepared oocytes were then washed in PBS before they were individually placed into compartments of 96-well PCR plates (Life Technologies, Austria) containing 10 µl of oocyte-lysis buffer (*Lee et al., 2012*) (50 mM Tris-HCl, [p.H 8.5], 1 mM EDTA, 0.5% tween-20 [Sigma–Aldrich, Austria] and 200 µg/ml Proteinase K [Macherey–Nagel, Germany]). Samples covered stages from primary oocytes of 3 day-old mice up to mature oocytes of 40 day-old mice. Samples were lysed at 55°C for 2 hr, and incubated at 95°C for 10 min to inactivate Proteinase K. The cellular DNA extract was finally diluted in 190 µl Tris-EDTA buffer, pH 8.0 (Sigma–Aldrich, Austria). 3 µl were used per qPCR reaction.

## Heteroplasmy quantification by Amplification Refractory Mutation System (ARMS)-qPCR

Heteroplasmy quantification was performed by ARMS-qPCR, an established method in the field (*Steinborn et al., 2000*; *Paull et al., 2013*; *Tachibana et al., 2013*), as described in *Burgstaller et al. (2014)*. The study was conducted according to MIQE (minimum information for publication of quantitative real-time PCR experiments) guidelines (*Bustin et al., 2009*; *Burgstaller et al., 2014*). The proportion between HB derived and C57BL/6N mtDNA was determined by ARMS-qPCR assays based on a SNP in mt-rnr2 (*Burgstaller et al., 2014*). These assays were normalised to changes in the input mtDNA amount by consensus assays, located in conserved regions of mt–Co2 and mt–Co3 (see Appendix 1). For the calculation of mtDNA heteroplasmy, the assay detecting the minor allele (C57BL/6N or wild-derived <50%) was always used. If both specific assays gave values >50% (which can happen around 50% heteroplasmy), the mean value of both assays was taken. All qPCR runs contained no template controls (NTCs) for all assays; these were negative in 100%. Further experimental details available in Appendix 1.

## Analytic model

In the BDP model, processes within a cell cycle constitute a birth-death process which can be solved using generating functions (*Gardiner, 1985*). For binomial partitioning, the generating function for the system after an arbitrary number of divisions has a recursive structure (*Rausenberger and Kollmann, 2008*; *Johnston and Jones, 2015*) and an analytic solution can be obtained through solving a Riccati recurrence relation. This reasoning also extends to the different phases of replication and degradation, allowing an exact generating function to be constructed for an arbitrary point in the bottleneck. Derivatives of this generating function are then used to obtain moments of the distributions of interest. The full procedure is given in Appendix 1. Recall that we assume that the bottlenecking process consists of a series of dynamic phases, which may either involve cycling cells (and hence cell divisions) or quiescent cells. The expression for mean mtDNA copy number $\mathbb{E}(m, t)$ at time $t$ is:

$$\mathbb{E}(m, t) = m_0 e^{\left(t - \tau^*\right)} \prod_{\text{phases } i} \frac{e^{(n_i \tau_i + \tau'_i)(\lambda_i - \nu_i)}}{2^{n_i}}, \tag{2}$$

where $n_i$ is the number of cell divisions in phase $i$ (0 for quiescent phases), $\tau_i$ is the length of a cell cycle in cycling phase $i$, $\tau'_i$ is the time spent in quiescent phase $i$ (0 for cycling phases), and $\tau^* = \Sigma_i(n_i \tau_i + \tau'_i)$, so that $t - \tau^*$ is the time since the last cell division. $\mathbb{E}(m, t)$ is thus intuitively interpretable as a product of the initial copy number with the effects of halving at each cell division, and the copy number evolution through past and current cell cycles and quiescent phases.

The expression for the variance is lengthier, taking the form

$$\mathbb{V}(m, t) = \frac{\Phi \mathbb{E}(m, t)}{\prod_{\text{phases } i} 4^{n_i} \left(e^{(\lambda_i - \nu_i)\tau_i} - 2\right)^2 (\lambda_i - \nu_i)^2} + \mathbb{E}(m, t) - \mathbb{E}(m, t)^2, \tag{3}$$

where $\Phi$ is a lengthy, though algebraically simple, function of all physical parameters, which we derive and present in Appendix 1. Once the means and variances associated with mutant and wild-type mtDNAs have been determined (for brevity, we write these as $\mu_1 \equiv \mathbb{E}(m_1, t), \sigma_1^2 \equiv \mathbb{V}(m_1, t)$ and $\mu_2 \equiv \mathbb{E}(m_2, t), \sigma_2^2 \equiv \mathbb{V}(m_2, t)$), the relations below can be used to compute heteroplasmy statistics:

$$\mathbb{E}(h) = \frac{\mu_2}{\mu_1 + \mu_2} \equiv \mu_h, \tag{4}$$

$$\mathbb{V}(h) = \mu_h^2 \left( \frac{\sigma_2^2}{\mu_2^2} - \frac{2\sigma_2^2}{\mu_2(\mu_1 + \mu_2)} + \frac{\sigma_1^2 + \sigma_2^2}{(\mu_1 + \mu_2)^2} \right). \tag{5}$$

## Selection

The predicted mean heteroplasmy at time $t$ assuming a constant selective pressure (though this assumption can straightforwardly be relaxed) is given by *Equation 4*, which, given *Equation 2*, straightforwardly reduces to

$$\mathbb{E}(h) = \frac{1}{1 + \frac{1 - h_0}{h_0} e^{-\Delta\lambda t}}, \tag{6}$$

where $h_0$ is initial heteroplasmy and $\Delta\lambda$ is the increase (or decrease, if negative) in replication rate of mutant over wild-type mtDNA. *Equation 6* predicts that mean heteroplasmy in the presence of selection will follow a sigmoidal form (as expected from population dynamics [*Futuyma, 1997*], by the constraint that $h_0$ must lie between 0 and 1, and by the intuitive fact that heteroplasmy changes slow down as these limits are approached).

## Threshold crossing

The probability of heteroplasmy exceeding a certain threshold $h^*$ is simply given by integrating the probability distribution of heteroplasmy between $h^*$ and 1. The exact distribution of heteroplasmy can be written as a sum over hypergeometric functions; however, for computational efficiency and interpretability, we employ an approximation to this distribution involving

the truncated Normal distribution and fixation probabilities. As shown in Appendix 1, the distribution of heteroplasmy, taking possible fixation into account, can be well approximated by

$$\mathbb{P}(h) = (1 - \zeta_1 - \zeta_2)\mathscr{N}'(h \mid \mu, \sigma^2) + \zeta_1\delta(h) + \zeta_2\delta(h-1), \tag{7}$$

where $\mathscr{N}'$ is the truncated Normal distribution (truncated at 0 and 1), $\mu$ and $\sigma^2$ are found numerically given our model results for $\mathbb{E}(h)$ and $\mathbb{V}(h)$, and $\zeta_1 \equiv \mathbb{P}(h=0)$ and $\zeta_2 \equiv \mathbb{P}(h=1)$ are fixation probabilities, also straightforwardly calculable from our model. The probability of threshold crossing for $0 < h^* < 1$ is then

$$\mathbb{P}(h > h^*) = (1 - \zeta_1 - \zeta_2)\left(1 - \frac{1}{2}\left(1 + \mathrm{erf}\left((h^* - \mathbb{E}(h))\big/\sqrt{2\mathbb{V}(h)}\right)\right)\right) + \zeta_2. \tag{8}$$

## Inference from heteroplasmy measurements

Given a sampled measurement heteroplasmy $h_m$, the probability $\mathbb{P}(h_0|h_m)$ that embryonic heteroplasmy is $h_0$ is given by Bayes' theorem $\mathbb{P}(h_0|h_m) = \mathbb{P}(h_m|h_0)\mathbb{P}(h_0)/\mathbb{P}(h_m)$. Assuming a uniform prior distribution on embryonic heteroplasmy (though this can be straightforwardly generalised), we thus obtain $\mathbb{P}(h_0|h_m) = \mathbb{P}(h_m|h_0)/\int_0^1 \mathbb{P}(h_m|h_0')dh_0'$, and using the above expression for the heteroplasmy,

$$\mathbb{P}(h_0 \mid h_m) = \frac{(1 - \zeta_1 - \zeta_2)\mathscr{N}'(h_m \mid \mu, \sigma^2) + \zeta_1\delta(h_m) + \zeta_2\delta(h_m - 1)}{\int_0^1 dh_0'(1 - \zeta_1 - \zeta_2)\mathscr{N}'(h_m \mid \mu, \sigma^2) + \zeta_1\delta(h_m) + \zeta_2\delta(h_m - 1)}, \tag{9}$$

where $\mu, \sigma^2, \zeta_1, \zeta_2$ are functions of $h_0$: $\mu, \sigma^2$ may be found numerically and the $\zeta$ values are analytically calculable (see Appendix 1).

## Additional information

### Funding

| Funder | Grant reference | Author |
|---|---|---|
| Medical Research Council (MRC) | MR/J013617/1 | Iain G Johnston |
| Biotechnology and Biological Sciences Research Council (BBSRC) | BB/D020190/1 | Nick S Jones |
| Medical Research Council (MRC) | MR/J010448/1 | Jo Poulton |
| Wellcome Trust | 0948685/Z/10/Z | Jo Poulton |

The funders had no role in study design, data collection and interpretation, or the decision to submit the work for publication.

### Author contributions

IGJ, Theoretical and experimental conception and design, Acquisition of data, Analysis and interpretation of data, Drafting or revising the article; JPB, GB, Experimental conception and design, Acquisition of data, Analysis and interpretation of data; VH, TK, TR, Acquisition of data, Analysis and interpretation of data; JP, Theoretical conception and design, Analysis and interpretation of data, Drafting or revising the article; NSJ, Theoretical and experimental conception and design, Analysis and interpretation of data, Drafting or revising the article

### Ethics

Animal experimentation: The study was discussed and approved by the institutional ethics committee in accordance with Good Scientific Practice (GSP) guidelines and national legislation. FELASA recommendations for the health monitoring of SPF mice were followed. Approved by the institutional ethics committee and the national authority according to Section 26 of the Law for Animal Experiments, Tierversuchsgesetz 2012 - TVG 2012.

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

# Appendix 1

## Data from experimental studies

*Table 1* contains the datapoints used in this study. These data are taken from *Tables 1*, *2* of *Cree et al. (2008)* (labelled 'Cao'); *Tables 1*, *2* of *Cao et al. (2007)*; *Appendix figure 1* of *Wai et al. (2008)* and *Appendix figure 1* of the following correspondence (reply to *Samuels et al., 2010*) (labelled 'Wai'); and *Table 2* of *Jenuth et al. (1996)* (labelled 'Jenuth'). Convention in the literature suggests that normalisation of measured heteroplasmy variance values, performed by division by a factor $h(1 - h)$, allows comparison of variance values from lines with diverse absolute heteroplasmies: the Wai data from correspondence is already normalised, and we manually normalised the Jenuth data using the $h$ values present.

**Table 1**. Source data used in this study

| Time/dpc | $\mathbb{E}(m)$ | $N$ | Source study |
|---|---|---|---|
| 0 | 2.5e5 | 22 | Cree† |
| 0.29 | 2.5e5 | 18 | Cree† |
| 0.58 | 5.8e4 | 9 | Cree† |
| 0.73 | 6.3e4 | 19 | Cree† |
| 0.88 | 4.8e4 | 33 | Cree† |
| 1.31 | 1.8e4 | 11 | Cree† |
| 7.5 | 4.5e2 | 596 | Cree‡ |
| 8.5 | 1.1e3 | 165 | Cree‡ |
| 10.5 | 1.6e3 | 96 | Cree‡ |
| 14.5 | 1.8e3 | 2615 | Cree‡, § |
| 0 | 1.7e5 | 42 | Cao† |
| 0.29 | 7.1e4 | 32 | Cao† |
| 0.58 | 4.8e4 | 32 | Cao† |
| 0.88 | 2.1e4 | 32 | Cao† |
| 5.5 | 1.5e3 | 85 | Cao‡, § |
| 6.5 | 1.6e3 | 43 | Cao‡, § |
| 7.5 | 2.0e3 | 53 | Cao‡, § |
| 7.75 | 2.0e3 | 42 | Cao‡, § |
| 8.5 | 2.1e3 | 82 | Cao‡, § |
| 9.5 | 2.2e3 | 93 | Cao‡, § |
| 10.5 | 2.1e3 | 74 | Cao‡, § |
| 11.5 | 2.0e3 | 67 | Cao‡, § |
| 12.5 | 1.9e3 | 124 | Cao‡, § |
| 13.5 | 2.1e3 | 71 | Cao‡, § |
| 8.5 | 2.8e2 | 20 | Wai# |
| 9.5 | 2.5e3 | 20 | Wai# |
| 10.5 | 2.8e3 | 20 | Wai# |
| 12.5 | 4.0e3 | 20 | Wai# |
| 14.5 | 5.8e3 | 20 | Wai# |
| 16.5 | 2.4e3 | 20 | Wai# |
| 25 | 5.0e3 | 20 | Wai# |
| 29 | 1.0e4 | 20 | Wai# |

*Table 1. Continued on next page*

Table 1. Continued

| Time/dpc | $\mathbb{E}(m)$ | N | Source study |
|---|---|---|---|
| 32 | 3.0e4 | 20 | Wai# |
| 46 | 6.5e4 | 20 | Wai# |

| Time/dpc | $\mathbb{V}'(h)$ | N | Source study |
|---|---|---|---|
| 7.5 | 5.2e-6 | 12 | Jenuth¶ |
| 7.5 | 0.008 | 4 | Jenuth¶ |
| 7.5 | 0.004 | 3 | Jenuth¶ |
| 7.5 | 0.008 | 5 | Jenuth¶ |
| 23 | 0.039 | 40 | Jenuth¶ |
| 23 | 0.032 | 37 | Jenuth¶ |
| 23 | 0.040 | 35 | Jenuth¶ |
| 23 | 0.038 | 35 | Jenuth¶ |
| 23 | 0.037 | 34 | Jenuth¶ |
| 23 | 0.021 | 48 | Jenuth¶ |
| 23 | 0.026 | 45 | Jenuth¶ |
| * | 0.140 | 26 | Jenuth¶, ** |
| * | 0.017 | 24 | Jenuth¶, ** |
| * | 0.144 | 31 | Jenuth¶, ** |
| * | 0.021 | 49 | Jenuth¶, ** |
| * | 0.033 | 31 | Jenuth¶, ** |
| 8.5 | 0.016 | 20 | Wai††, ‡‡ |
| 8.5 | 0.018 | 20 | Wai††, ‡‡ |
| 9.5 | 0.018 | 20 | Wai††, ‡‡ |
| 10.5 | 0.017 | 20 | Wai††, ‡‡ |
| 10.5 | 0.025 | 20 | Wai††, ‡‡ |
| 10.5 | 0.008 | 20 | Wai††, ‡‡ |
| 12.5 | 0.017 | 20 | Wai††, ‡‡ |
| 13.5 | 0.014 | 20 | Wai††, ‡‡ |
| 13.5 | 0.015 | 20 | Wai††, ‡‡ |
| 13.5 | 0.020 | 20 | Wai††, ‡‡ |
| 13.5 | 0.021 | 20 | Wai††, ‡‡ |
| 14.5 | 0.004 | 20 | Wai††, ‡‡ |
| 16.5 | 0.015 | 20 | Wai††, ‡‡ |
| 25.0 | 0.002 | 20 | Wai††, ‡‡ |
| 25.0 | 0.002 | 20 | Wai††, ‡‡ |
| 29.0 | 0.013 | 20 | Wai††, ‡‡ |
| 29.0 | 0.027 | 20 | Wai††, ‡‡ |
| 32.0 | 0.016 | 20 | Wai††, ‡‡ |
| 32.0 | 0.023 | 20 | Wai††, ‡‡ |
| 35.0 | 0.026 | 20 | Wai††, ‡‡ |
| 35.0 | 0.029 | 20 | Wai††, ‡‡ |
| 50.0 | 0.039 | 20 | Wai††, ‡‡ |
| 51.1 | 0.043 | 20 | Wai††, ‡‡ |
| 65.0 | 0.026 | 20 | Wai††, ‡‡ |

†Data referenced by number of cells post–conception is assigned a time measurement assuming the 7 hr → 16 hr doubling times from *Lawson and Hage (1994)*.

‡Mean copy number taken directly from tabulated data.

§(Weighted) average over germline cell classes presented at this time point.

#Extracted from data in figures; *n* not explicitly available so estimated as *n* = 20 from accompanying histograms and discussion.

¶Manually normalised from given data.

**Data from mature oocytes in next generation: time in dpc not available.

††Extracted from data in figure in correspondence following study.

‡‡*n* not explicitly available so estimated as *n* = 20 from accompanying histograms and discussion in original paper.

**Table 2**. New heteroplasmy measurements from the HB model system

| Age | 3 | 3 | 4 | 4 | 4 | 4 | 8 | 8 | 9 | 9 | 9 | 24 | 37 | 37 | 40 |
| *n* | 25 | 30 | 21 | 13 | 13 | 11 | 30 | 34 | 20 | 17 | 36 | 25 | 24 | 20 | 20 |
|---|---|---|---|---|---|---|---|---|---|---|---|---|---|---|---|
| $\mathbb{E}(h)$ | 0.501 | 0.419 | 0.183 | 0.337 | 0.382 | 0.354 | 0.301 | 0.559 | 0.193 | 0.245 | 0.049 | 0.457 | 0.566 | 0.276 | 0.238 |
| $\mathbb{V}(h)$ | 0.00256 | 0.00359 | 0.00824 | 0.01308 | 0.00913 | 0.00461 | 0.00350 | 0.00631 | 0.00408 | 0.00301 | 0.00097 | 0.00625 | 0.01000 | 0.00913 | 0.00662 |
| $\mathbb{V}'(h)$ | 0.0102 | 0.0147 | 0.0551 | 0.0585 | 0.0387 | 0.0202 | 0.0167 | 0.0256 | 0.0262 | 0.0163 | 0.0210 | 0.0252 | 0.0407 | 0.0457 | 0.0364 |
| $h \times 100$ | 40.8 | 26.5 | 7.4 | 16.7 | 26.9 | 25.1 | 18.2 | 38.7 | 8.3 | 13.3 | 1.7 | 32.5 | 38.0 | 12.8 | 8.9 |
| | 43.4 | 30.6 | 7.6 | 23.1 | 28.8 | 25.7 | 22.0 | 41.7 | 9.3 | 16.9 | 1.7 | 32.6 | 39.2 | 13.8 | 12.1 |
| | 44.1 | 31.8 | 7.9 | 24.0 | 29.4 | 29.1 | 23.7 | 43.9 | 10.8 | 20.1 | 1.8 | 37.1 | 45.9 | 15.9 | 13.9 |
| | 44.2 | 33.2 | 9.6 | 24.4 | 29.8 | 30.8 | 23.9 | 46.4 | 13.5 | 20.2 | 1.9 | 37.1 | 50.9 | 16.6 | 15.7 |
| | 46.6 | 36.4 | 9.7 | 26.8 | 30.2 | 36.5 | 24.6 | 47.5 | 13.5 | 20.2 | 2.0 | 38.0 | 51.0 | 20.3 | 18.8 |
| | 46.7 | 37.7 | 10.3 | 27.0 | 36.7 | 36.7 | 25.9 | 47.9 | 15.8 | 21.3 | 2.8 | 39.5 | 51.7 | 20.6 | 19.1 |
| | 46.9 | 37.9 | 12.4 | 28.6 | 37.1 | 38.2 | 26.0 | 49.5 | 16.3 | 23.7 | 2.8 | 40.0 | 51.7 | 21.8 | 19.5 |
| | 47.4 | 38.7 | 14.3 | 39.8 | 40.2 | 38.3 | 26.2 | 51.4 | 16.4 | 24.9 | 2.9 | 41.1 | 51.8 | 23.6 | 20.8 |
| | 48.0 | 39.2 | 14.8 | 40.4 | 40.5 | 40.4 | 26.3 | 51.5 | 18.4 | 25.2 | 2.9 | 43.0 | 52.0 | 23.8 | 22.5 |
| | 48.4 | 39.5 | 16.0 | 41.9 | 43.6 | 43.6 | 26.9 | 52.8 | 18.7 | 26.2 | 3.0 | 43.7 | 54.1 | 26.2 | 22.7 |
| | 48.5 | 39.7 | 17.0 | 42.9 | 45.8 | 44.8 | 26.9 | 52.8 | 20.7 | 27.0 | 3.0 | 43.7 | 54.2 | 28.3 | 23.2 |
| | 48.7 | 41.4 | 17.1 | 50.1 | 46.7 | – | 27.8 | 52.9 | 20.7 | 27.3 | 3.0 | 44.2 | 54.4 | 29.6 | 23.4 |
| | 49.3 | 42.1 | 18.6 | 52.8 | 60.5 | – | 28.9 | 53.2 | 21.3 | 27.6 | 3.0 | 44.5 | 54.9 | 32.6 | 27.8 |
| | 50.3 | 42.6 | 19.7 | – | – | – | 29.1 | 53.2 | 21.8 | 27.8 | 3.3 | 46.3 | 55.3 | 33.1 | 28.9 |
| | 50.5 | 42.7 | 20.8 | – | – | – | 29.7 | 53.5 | 23.8 | 28.2 | 3.3 | 46.6 | 55.7 | 33.1 | 29.4 |
| | 50.6 | 42.7 | 25.7 | – | – | – | 29.9 | 53.9 | 24.2 | 30.0 | 3.5 | 47.0 | 57.2 | 34.5 | 30.1 |
| | 50.8 | 42.9 | 25.7 | – | – | – | 30.1 | 54.2 | 26.5 | 36.7 | 3.5 | 49.1 | 57.5 | 38.4 | 30.9 |
| | 51.2 | 43.8 | 26.4 | – | – | – | 30.7 | 55.2 | 26.5 | – | 3.7 | 49.8 | 59.9 | 40.5 | 34.8 |
| | 53.7 | 44.5 | 28.3 | – | – | – | 31.3 | 56.0 | 26.8 | – | 3.8 | 50.1 | 61.1 | 42.2 | 35.2 |
| | 54.5 | 44.7 | 35.6 | – | – | – | 31.7 | 56.2 | 32.1 | – | 3.8 | 51.8 | 64.8 | 43.9 | 39.4 |
| | 55.0 | 44.8 | 39.3 | – | – | – | 32.6 | 57.1 | – | – | 4.0 | 53.0 | 69.9 | – | – |
| | 56.2 | 45.9 | – | – | – | – | 33.0 | 57.3 | – | – | 4.0 | 54.3 | 74.1 | – | – |
| | 56.6 | 47.0 | – | – | – | – | 33.4 | 59.8 | – | – | 4.9 | 55.7 | 76.1 | – | – |
| | 59.0 | 47.6 | – | – | – | – | 33.6 | 60.0 | – | – | 5.5 | 55.7 | 76.5 | – | – |
| | 62.0 | 48.7 | – | – | – | – | 34.7 | 60.1 | – | – | 5.8 | 66.2 | – | – | – |
| | – | 48.7 | – | – | – | – | 34.9 | 61.3 | – | – | 5.9 | – | – | – | – |
| | – | 48.8 | – | – | – | – | 35.3 | 61.3 | – | – | 6.0 | – | – | – | – |
| | – | 48.9 | – | – | – | – | 35.8 | 62.1 | – | – | 6.1 | – | – | – | – |
| | – | 49.1 | – | – | – | – | 41.2 | 65.6 | – | – | 6.7 | – | – | – | – |
| | – | 49.7 | – | – | – | – | 48.5 | 67.1 | – | – | 7.9 | – | – | – | – |
| | – | – | – | – | – | – | – | 68.3 | – | – | 8.2 | – | – | – | – |
| | – | – | – | – | – | – | – | 69.2 | – | – | 8.3 | – | – | – | – |
| | – | – | – | – | – | – | – | 69.4 | – | – | 8.5 | – | – | – | – |
| | – | – | – | – | – | – | – | 70.1 | – | – | 8.8 | – | – | – | – |
| | – | – | – | – | – | – | – | – | – | – | 11.6 | – | – | – | – |
| | – | – | – | – | – | – | – | – | – | – | 16.6 | – | – | – | – |

Heteroplasmy measurements and statistics from the HB model system. Ages are given in days after birth.

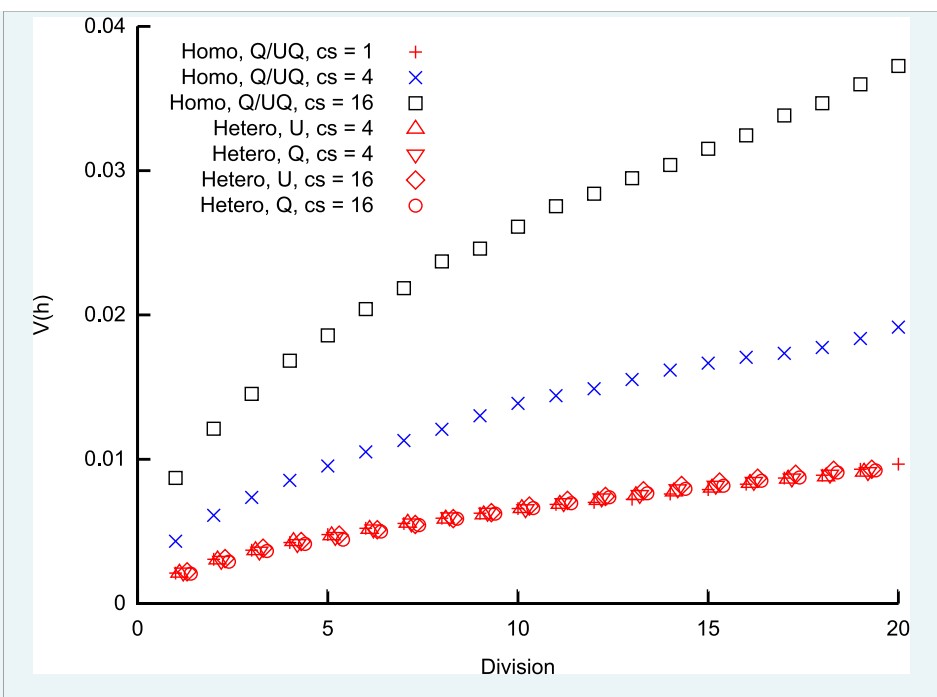

**Appendix figure 1**. Heteroplasmy variance in a model system under several different group-inheritance regimes. $\mathbb{V}(h)$ over many cell divisions when the elements of inheritance are heteroplasmic or homoplasmic groups of different size. Groups may be quenched (Q; constituents remain the same across cell divisions) or unquenched (UQ; constituents are randomly resampled from the cellular population each cell cycle); for homoplasmic clusters, an unquenched protocol yields identical results to the quenched protocol. $\mathbb{V}(h)$ behaviour differing from binomial partitioning ($c = 1$) is only observed for homoplasmic groups with $c \geq 2$. Points for heteroplasmic groups are slightly offset in the x-direction for clarity.

Where data in the original studies were presented as a function of number of cells in a developing organism, as opposed to an explicit function of time, we have assigned times using the 7 hr → 16 hr doubling times from *Lawson and Hage (1994)*. Other sources assume a 15 hr doubling time throughout early development: using the data interpreted in this way did not lead to a qualitative difference in our conclusions and very little quantitative change in posterior distributions (data not shown). Some datapoints did not have associated or readily available sample sizes $N$: for these datapoints we estimated $N$ using available evidence in the publication. To check for dependence on these values of $N$ we performed our inference process with a range of alternative $N$ values and with a test case where $N$ was set to 100 for every datapoint: all results and posteriors were qualitatively similar, showing a lack of strong dependence of our conclusions on the specific numbers of samples involved in deriving the experimental measurements (data not shown).

## Heteroplasmic and homoplasmic clusters

The specific units of inheritance of mtDNA have been debated in the literature for decades. The smallest possible unit of inheritance is a single mtDNA molecule; some studies have hypothesised that the unit of inheritance consists of groups of mtDNA molecules. Within this picture, debate exists as to whether these groups are semi-permanent associations of molecules (which we will refer to as 'quenched' sets) or more fluid transient colocalisations of molecules (which we refer to as 'unquenched'). Furthermore, the size of these units is debated, with estimates ranging from an average size of 1.4–10 mtDNA molecules (*Bogenhagen, 2012*; *Kukat and Larsson, 2013*), and it is unknown whether the mtDNAs within a group are strictly homoplasmic or if heteroplasmic groups are possible, although current evidence, at the finest resolution, points towards homoplasmic groups of size <2 (*Gilkerson et al., 2008*; *Poe et al., 2010*; *Wallace and Chalkia, 2013*; *Jakobs and Wurm, 2014*).

We will classify these different pictures with three parameters. First, the characteristic size $c$ of an mtDNA group. Second, a classifier denoting whether these groups are quenched (in the sense that the individual constituents of a group remain the same over many cell divisions) or unquenched (in the sense that the individual constituents of a group may change between cell divisions). Third, a classifier denoting whether groups are necessarily homoplasmic, or if heteroplasmy is permitted.

An early hypothesis from *Jacobs et al. (2000)* considered 'nucleoids' which correspond to quenched heteroplasmic groups with $c > 1$, retaining their internal structure across cell divisions and containing different mtDNA types. If the mitochondrial organelle is the unit of inheritance, we may expect unquenched heteroplasmic groups with $c > 1$, as mitochondrial dynamics act to mix the content of the mitochondrial system between cell divisions, but organelles are likely to contain more than one mtDNA molecule. If nucleoids are the units of inheritance and, as current understanding suggests, nucleoids are small and homoplasmic (if mtDNA indeed exists in groups at all), the appropriate picture is $c \simeq 1$, homoplasmic groups.

Here we show that the heteroplasmy statistics resulting from these different pictures of grouped inheritance collapse onto two representative cases: first, that corresponding to homoplasmic clusters with $c > 1$, and second, that corresponding to $c = 1$ (binomial inheritance). Quenching—whether mtDNA content can remix within nucleoids—is shown to be unimportant in determining heteroplasmy statistics. Our model for these different situations is as follows. We consider a cellular population as consisting of a set of mtDNA molecules, existing in groups of size $c$. During a cell cycle, the population of groups doubles deterministically (we ignore random birth-death dynamics in this model, in order to focus on partitioning dynamics), so that every group produces one exact copy of itself. For unquenched simulations, a new set of groups is then formed by resampling the individual mtDNA constituents of the cell. For quenched simulations (representing the situation postulated in *Jacobs et al. (2000)*), the existing groups remain intact. At cell divisions, groups are binomially partitioned between the two daughter cells.

The model is initialised with a cell containing $m_0$ mtDNAs, split into $(1 - h)m_0$ wild-type and $hm_0$ mutant molecules. These mtDNAs are clustered into $m_0/c$ groups, according to the cluster picture under consideration (i.e., homoplasmic or heteroplasmic clusters). We simulate the subsequent doubling then partitioning of this system through cell divisions many times, assuming a constant cell cycle length, and record the cell-to-cell heteroplasmy variance with time.

*Appendix figure 1* shows the resultant heteroplasmy variance trajectories for different cases (with $h_0 = 0.1$; other initial heteroplasmies showed similar behaviour). The first striking result is that the inheritance of heteroplasmic groups produces the same heteroplasmy variance as binomial partitioning, regardless of cluster size. This behaviour is due to the balance between stochasticity associated with the makeup of, and partitioning of, groups. A small number of large groups will experience substantial partitioning noise, but larger heteroplasmic groups are more likely to faithfully represent the overall cell heteroplasmy. As identified in *Jacobs et al. (2000)*, the inheritance of heteroplasmic groups thus provides a means to facilitate local mtDNA complementation while provoking no increase in heteroplasmy variance beyond that associated with binomial partitioning of elements at divisions.

We also observe that quenched populations behave in the same way as unquenched populations. In the case of homoplasmic groups, this result is obvious, as a set of homoplasmic nucleoids of a given size can only be constructed in one way for a given number of mtDNA molecules of different types. For heteroplasmic groups, this result implies that resampling the cellular population to produce a new group produces a negligible amount of additional stochasticity compared to that already present in the random makeup and inheritance of groups. Thus, the only determinant factors of heteroplasmy variance related to the inheritance of groups are whether groups are homoplasmic or heteroplasmic, and, if the former, the characteristic size of groups.

These results illustrate that the binomial inheritance model can also describe the statistics associated with heteroplasmic nucleoids of arbitrary size, over a timescale of several dozen cell divisions (suitable to describe the developmental process). The theoretical long-term behaviour of these systems involves some more subtleties. At much longer times, the probability that all mtDNA types become extinct in a cell is not negligible. When complete extinction cannot be ignored, heteroplasmy statistics become poorly defined. This extinction timescale is shorter for cluster inheritance than for binomial inheritance, as a greater variability in copy number (though not in heteroplasmy) results from each division for larger clusters. However, our simulations indicate that as long as the heteroplasmy variance associated with heteroplasmic clusters remains well defined, it matches that resulting from binomial inheritance.

We propose that a reasonable view may be that individual mitochondrial fragments, including several small, homoplasmic nucleoids, are the likely elements of inheritance at partitioning. Furthermore, there is likely some movement of these nucleoids within the mitochondrial network, and fission and fusion likely mean that a given mtDNA will not be associated with the same static mitochondrial element in perpetuity. In this case, the picture of an unquenched, heteroplasmic group of mtDNAs—those contained within a discrete element of the mitochondrial system—seems most reasonable. We can thus speculate that, as demonstrated by the previous results, the precise size of mitochondrial fragments at partitioning is not important for the heteroplasmy dynamics (nor indeed is whether they are quenched or unquenched). Our simple binomial partitioning model is thus consistent with what one might consider the most physiologically plausible model, and indeed with any models not involving large and strictly homoplasmic groups as the elements of mitochondrial inheritance.

## Parametric inference for bottlenecking dynamics

Our model is a function of the parameter set $\theta = \{\nu_i, \lambda_i, n_i, \tau_i, S, \alpha, T, c, h_0, m_0, \delta\lambda\}$. For reference, the meanings of these parameters are (as in **Figure 1D** in the Main text): replication ($\lambda_i$) and degradation ($\nu_i$) rates; number of cell divisions ($n_i$) and cell cycle length ($\tau_i$) in each dynamic phase $i$; deterministic or stochastic dynamics label ($S = 0, 1$ respectively); a proportion $\alpha$ of mtDNAs capable of replication after threshold time $T$; deterministic ($c = 0$), binomial ($c = 1$) or clustered ($c > 1$) partitioning at divisions; initial heteroplasmy $h_0$ and initial copy number $m_0$; $\delta\lambda$ is an additional parameter allowing a possible difference in replicative rates between mutant and wildtype mtDNA: this is zero unless otherwise stated. For the following parameters we use uninformative uniform priors on the given interval: $\lambda_i, \nu_i \in [0,1]$ hr$^{-1}$; $S \in \{0,1\}$; $\alpha \in [0.005,1]$; $T \in [0,100]$ day; $c \in [0,100]$ day; $h_0 \in [0,1]$; $m_0 \in [0,10^6]$. The following values are fixed from experimental studies (**Lawson and Hage, 1994**; **Drost and Lee, 1995**): $n_1 = 29$; $n_2 = 7$; $\tau_1 = 7$ hr; $\tau_2 = 16$ hr. $\lambda_6 = 0$ hr$^{-1}$ is fixed to avoid mtDNA proliferation after development; $h_0 = 0.2$ is fixed as an intermediate value as heteroplasmy variance measurements are generally normalised; $\delta\lambda$, a parameter allowing a difference in replicative rates between mutant and wildtype mtDNAs, is fixed at zero throughout as we ignore selective pressure. The parameter $\tau_i$ for $i > 2$ is used to determine the length of time spent in different quiescent phases and is subject to the uniform prior $\tau_i \in [0, 50]$ day.

Given these priors, we use an approximate Bayesian computation (ABC) approach to build a posterior distribution over the parameters in our bottlenecking model (**Toni et al., 2009**). ABC involves using a summary statistic $\rho(\theta, \mathscr{D})$ to compare the available data $\mathscr{D}$ to the predictions of a model given parameters $\theta$. If parameter sets are sampled from the set for which $\rho \leq \epsilon$, where $\epsilon$ is a threshold difference between the resulting model behaviour and experimental data, the posterior distribution $P(\theta|\rho(\theta, \mathscr{D} \leq \epsilon))$ is sampled, which is argued to sufficiently approximate $P(\theta|\mathscr{D})$ for suitably small $\epsilon$ (**Marjoram et al., 2003**).

We define a residual sum-of-squares difference between the results of a simulated model and experimental data (**Johnston, 2014**):

$$\rho(\theta, \mathscr{D}) = \sum_{i=0}^{N_m} \left( \log \mathbb{E}_\theta \left( m \mid t = t^{(i)}_{\mathscr{D},m} \right) - \log \mathbb{E}_\mathscr{D} \left( m \mid t = t^{(i)}_{\mathscr{D},m} \right) \right)^2 + \sum_{i=0}^{N_h} A_1 \left( \mathbb{V}_\theta \left( h \mid t = t^{(i)}_{\mathscr{D},h} \right) - \mathbb{V}_\mathscr{D} \left( h \mid t^{(i)}_{\mathscr{D},h} \right) \right)^2,$$

(1)

where $\mathscr{D}$ denotes experimental data. We thus amalgamate experimental results of two types: mean mtDNA copy number (with $N_m$ data points measuring $\mathbb{E}_\mathscr{D}(m)$ at times $t^{(i)}_{\mathscr{D},m}$), and mean and variance of heteroplasmy (with $N_h$ data points measuring $\mathbb{V}_\mathscr{D}(h)$ at times $t^{(i)}_{\mathscr{D},h}$). The sets of data for $\mathbb{E}(m)$ and $\mathbb{V}(h)$ contain different numbers of points and are of different absolute magnitudes. We compensate for these differences by using the logarithms of copy number measurements (as these values span several orders of magnitude), and weighting parameter $A_1 = 10^3$. This weighting parameter compensates for the different magnitudes and number of datapoints in each class of measurement, ensuring that the contribution to the total residual from each set of data is of comparable magnitude. Our summary statistic thus records a residual sum-of-squares difference between experiment and simulation values for $\log \mathbb{E}(m)$ and $\mathbb{V}(h)$ at each time point where an experimental measurement exists.

We performed our model selection process using several different alternative protocols, including comparing logarithms of $\mathbb{V}(h)$ measurements (in contrast to the raw values) and varying $A_1$ over orders of magnitude from $10^2$–$10^4$ (corresponding to unbalanced weighting, favouring $\mathbb{E}(m)$ and $\mathbb{V}(h)$ data respectively). In all cases, the BDP model identified in the Main text experienced substantially more support than any alternative. For inference involving the new dataset from the HB model system, we use the default protocol above and set $A_1 = 3 \times 10^3$ to account for the threefold decrease in available $\mathbb{V}'(h)$ datapoints.

We use an MCMC implementation of ABC, whereby we construct a Markov chain $\theta_i$, where each state consists of a set of trial parameters to be assessed. We create $\theta_{i+1}$ by perturbing each parameter within $\theta_i$ with a perturbation kernel consisting of a Normal distribution on each parameter with standard deviations between 0.1–1% of the width of the prior (varied as the model depends more sensitively on some parameters than others). In the case of discrete parameter $c$, a continuous representation $c'$ is used and varied in the MCMC approach, with $c = \lfloor 100c' \rfloor$. We accept $\theta_{i+1}$ as the new state of the chain if $\rho(\theta_{i+1}, \mathscr{D}) \leq \epsilon$. We ran $10^6$ MCMC iterations for ABC model selections and checked convergence by running five instances of each simulation for different random number seeds.

For the initial optimisation of model fitting, we ran $10^6$ MCMC steps using the protocol above but accepting a move according to the Metropolis–Hastings protocol (**Hastings, 1970**), recording the parameterisation leading to the lowest recorded residual. In this case we used uninformative initial conditions, with identical choices for all rate parameters, corresponding to an inaccurate trajectory of copy number and heteroplasmy variance. For model selection, we used the protocol above, with a different set of parameters $\theta^M$ for each model $M$, with each MCMC step proposing a random model from the Cao alone, Wai alone, and BDP set described in the text, as in the SMC ABC model selection protocol proposed in **Toni et al. (2009)**. We record the proportion of accepted steps involving each model type. The parameterisations found through initial optimisation were used as initial conditions in the ABC model selection and inference simulations.

Initial optimisation identified parameterisations all displaying residuals under $\epsilon = 50$. We chose $\epsilon = \{45, 50, 60, 75\}$ for the ABC model selection simulations to display the varying degrees of support for each model as stricter agreement with experiment was enforced. We chose $\epsilon = 40$ for the ABC inference of BDP model parameterisation to ensure these models all displayed better fits to data than the alternative models. In **Appendix figure 2** we illustrate the distribution of squared residuals for the BDP model under a range of $\epsilon$ values.

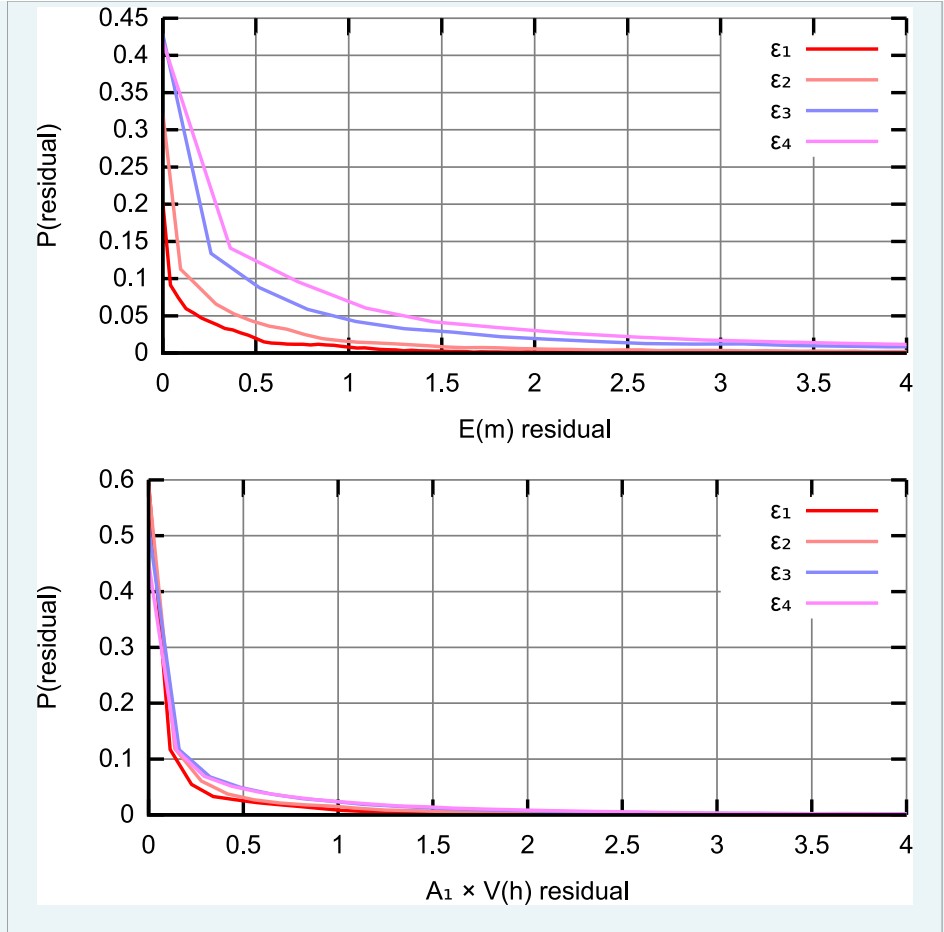

**Appendix figure 2**. Residual distributions at different ABC thresholds $\epsilon$. The distribution of squared residuals corresponding to individual experimental datapoints compared to an ensemble of simulated trajectories for (top) $\log\mathbb{E}(m)$ (bottom) $\mathbb{V}(h)$. The $\mathbb{V}(h)$ residuals are scaled by $A_1 = 10^3$ to ensure that the two sets of measurements are compared on a quantitatively equal footing. As $\epsilon$ is decreased ($\epsilon_{1,2,3,4} = 40$, 50, 75, 100), distributions of residuals from accepted trajectories tighten around zero.

## Posteriors for all variables and datasets

In *Appendix figure 3* we display all posterior distributions for all parameters resulting from our ABC approach assuming the BDP model. There is substantial variability in the possible timescales and magnitudes of random turnover associated with each random dynamic phase $i > 2$, exemplified by the complicated and bimodal structure of the posteriors on these parameters. This variability reflects the fact that an increase in heteroplasmy variance can be achieved through a variety of specific mtDNA trajectories, and current experimental data is insufficient to distinguish specific time behaviours within this variety. However, the total contribution of each random phase to the overall dynamics is more constrained, as shown in the posterior distribution on a measure of total random turnover $\sigma = \sum_{i=3}^{6} \tau'_i \nu_i$. This quantity is the sum over all later phases of the product of the length of that phase and the rate of random turnover, thus giving a measure of total random turnover. The fact that this posterior is more tightly constrained than the posteriors on individual $t_i$, $\nu_i$ parameters suggests that the required mtDNA turnover can be achieved through a range of specific dynamic trajectories from the inferred mechanism: for example, the exact time at which random mtDNA turnover sharply increases is currently flexible (though constrained to lie around 25 dpc) without more detailed data. This flexibility is also observed in the trajectories of posterior distributions in the Main text.

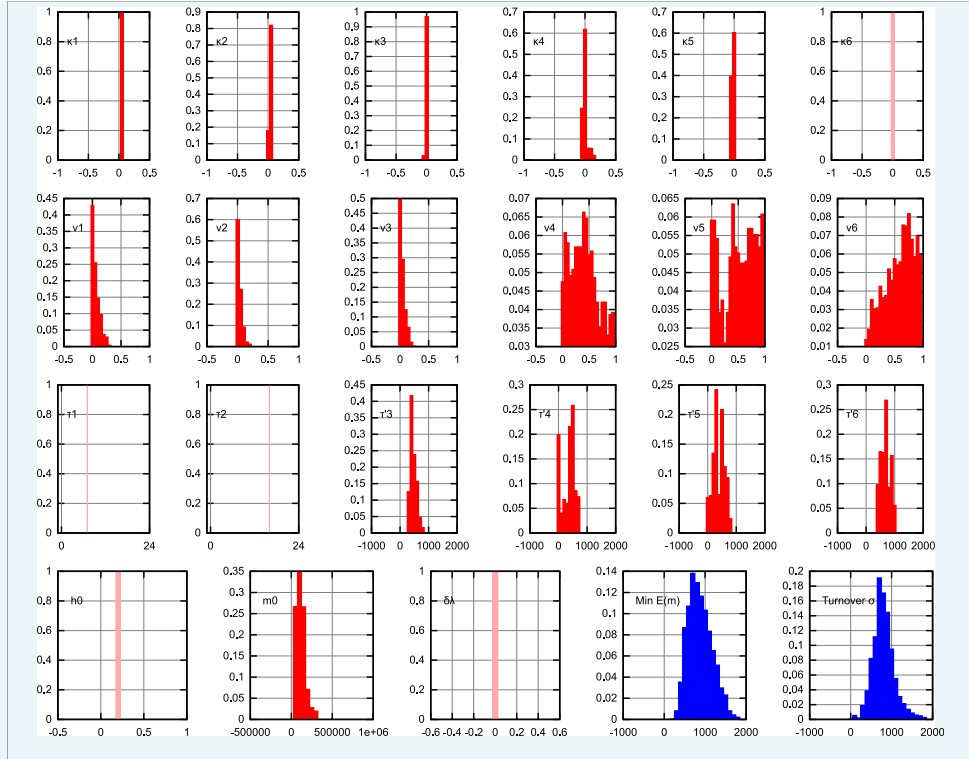

**Appendix figure 3**. Posterior distributions on model parameters. The posterior distributions on individual model parameters, assuming the inferred BDP bottlenecking mechanism. Replication rates are presented as $\kappa = \lambda - \nu$, thus representing overall proliferation rates of mtDNA. Units are omitted for clarity. Pale, single-values distributions correspond to parameter values fixed within the model ($\kappa_6 = 0$ to prevent mtDNA proliferation after development; $\tau_1 = 7$ hr, $\tau_2 = 16$ hr fixed by data on cell doubling times; $h_0 = 0.2$ fixed for simplicity as heteroplasmy variances are normalised; $\delta\lambda = 0$ fixed to avoid varying selective pressure). The 'turnover' parameter, described in the text, is $\sum_{i=3}^{6} \tau'_i \nu_i$, a measure of the total random turnover in the mtDNA population.

## Experimental measurements

*Table 2* contains the measurements of heteroplasmy $h$, mean heteroplasmy $\mathbb{E}(h)$, raw and normalised heteroplasmy variance $\mathbb{V}(h)$ and $\mathbb{V}'(h)$, and number of datapoints $n$, from the HB model system. Experimental procedures are described in 'Materials and methods'; further specifics follow.

Consensus assays:

Co2-f: GCCAATAGAACTTCCAATCCGTATAT,

Co2-r: TGGTCGGTTTGATGTTACTGTTG,

Co2-FAM: CTGATGCCATCCCAGGCCGACTAA-BHQ1 (Amplicon length: 136 bp);

Co3-f: TCTTATATGGCCTACCCATTCCAA,

Co3-r: GGAAAACAATTATTAGTGTGTGATCATG,

Co3-FAM: TTGGTCTACAAGACGCCACATCCCCT-BHQ1 (Amplicon length: 103 bp).

ARMS-assays:

16SrRNA2340(3)G-f: AATCAACATATCTTATTGACCaAG (haplotype C57BL/6N),

16SrRNA2340(3)A-f: AATCAACATATCTTATTGACCgAA (haplotype HB);

16SrRNA2458-r: CAC CAT TGG GAT GTC CTG ATC,

16SrRNA-FAM: FAM-CAA TTA GGG TTT ACG ACC TCG ATG TT-BHQ-1. (Amplicon length: 142 bp).

Every qPCR run consisted of one consensus and an ARMS assay.

Master-mixes for triplicate qPCR reactions contained 1× buffer B (Solis BioDyne, Estonia); 4.5 MgCl$_2$ for the ARMS and the Co3 consensus assays, and 3.5 mM MgCl$_2$ for the Co2 consensus assay; 200 M of each of the four deoxynucleotides (dNTPs, Solis BioDyne, Estonia), HOT FIREPol DNA polymerase according to the manufacturers instructions (Solis BioDyne, Estonia), 300 nM of each primer and 100 nM hydroloysis probe (Sigma–Aldrich, Austria). Per reaction 12 μl of master-mix and 3 μl DNA were transferred in triplicates to 384-well PCR plates (Life Technologies, Austria) using the automated pipetting system epMotion 5075TMX (Eppendorf, Germany). Amplification was performed on the ViiA 7 Real-Time PCR System using the ViiA 7 Software v1.1 (Life Technologies, USA). DNA denaturation and enzyme activation were performed for 15 min at 95°C. DNA was amplified over 40 cycles consisting of 95°C for 20 s, 58°C for 20 s and 72°C for 40 s for both assays.

The standard curve method was applied. Amplification efficiencies were determined for each run separately by DNA dilution series consisting of DNA from wild-derived mice, harbouring the respective analysed mtDNA. Typical results: slope = −3.665, −3.562, −3.461, −3.576; mean efficiency = 0.87, 0.9, 0.94, 0.90; and Y-intercept = 32.2, 28.3, 33.8, 34.5; for the consensus Co2, consensus Co3, C57BL/6N and HB assays respectively. Coefficient of correlation was >0.99 in all assays in all runs. All target samples lay within the linear interval of the standard curves. To test for specificity, in each run a negative control sample, that is, a DNA sample of a mouse harbouring the mtDNA of the non-analysed type in the heteroplasmic mouse (i.e., C57BL/6N or HB mtDNA) was measured. All assays could discriminate between C57BL/6N and HB mtDNA at a minimum level of 0.2%. Target sample DNA was tested for inhibition by dilution in Tris-EDTA buffer (Sigma–Aldrich, Austria), pH 8.0.

## Bottlenecking mechanisms and further experimental elucidation

Here we summarise potential mechanisms for the bottleneck that conflict with our statistical interpretation, highlighting the reasons for the conflict. We also propose further experiments that would efficiently provide more evidence to distinguish these hypotheses.

### Other proposed mechanisms

### Random partitioning of homoplasmic mtDNA clusters

*Cao et al. (2007)* suggests a less powerful depletion of mtDNA copy number during early development than assumed by other studies, with heteroplasmy variance increase instead being explained by the partitioning of clusters of mtDNA at cell divisions. However, the time period over which *Wai et al. (2008)* and *Jenuth et al. (1996)* observe increasing heteroplasmy variance corresponds to a situation in which germ line cells are largely quiescent, immediately suggesting that partitioning at cell divisions cannot explain increasing variance (as cell divisions do not occur). Furthermore, results from our model suggest that, unless these clusters are very small, this mechanism would immediately lead to a rather higher and sharper increase in heteroplasmy variance than observed.

### Replication of a specific subset of mtDNAs during folliculogenesis

*Wai et al. (2008)* proposes a mechanism in which only a subset of mtDNAs replicate during folliculogenesis. There are several specific dynamic schemes by which this mechanism could be manifest. The first that we consider involves the following scenario: at some point during development, around the start of folliculogenesis, a specific subset of mtDNAs in each cell is 'marked' as able to replicate (we next consider the case in which this subset is more plastic with time). In this case, the effect of 'switching off' replication of a subset of mtDNAs depends on the balance of replication and degradation rates of the mtDNA population:

- Low replication, low degradation. In this case, the population stays largely static; the switching off of replication has little effect, and the heteroplasmy variance cannot increase to the levels observed in experiment.
- Low replication, high degradation. In this case, the high degradation rate ensures that the non-replicating mtDNAs are removed from the cell, providing a 'bottleneck' as only the replicating mtDNAs remain. However, this regime yields a transient period of mtDNA copy number depletion, while the non-replicating mtDNAs are degrading but the (small) population of replicating agents remains low. This copy number depletion is not observed.
- High replication, high degradation. In this case, non-replicating mtDNAs are removed and replicating mtDNAs are capable of fast enough replication to survive the transient drop in copy number. However, the rates associated with this mechanism are necessarily high enough such that the increase in heteroplasmy variance is very sharp, notably more so than the smooth increase with time observed in experiment (see, e.g., *Figure 2* in the Main text).
- Combined subset replication, and/or heteroplasmic cluster inheritance, with and random dynamics. Our approach does not provide support against a model combining our inferred mechanism (increased random turnover of mtDNAs) with some other dynamic schemes, namely (a) that in which only a subset of mtDNAs may replicate during folliculogenesis, and/or (b) where heteroplasmic mtDNA clusters, rather than individual mtDNAs, are the units of inheritance. A combination with (a) would allow the reduction of the key parameters associated with each: so the rate of random turnover could be lower, and the proportion of replicating genomes larger, than in the case of the pure incarnations of those respective mechanisms. This scheme may thus provide a viable alternative—however, it requires an introduction of two coupled mechanisms, which experimental data currently cannot disambiguate. For this reason and for parsimony, we report the case where random dynamics alone are responsible, and below suggest experimental protocols to further elucidate this possible link or its absence. A combination with (b) is possible and cannot be discounted using the available data, as the trajectories of heteroplasmy variance under (b) and under binomial inheritance are the same. We propose observations of mitochondrial ultrastructure and mtDNA localisation during development to resolve this remaining mechanistic question.

## Observation of a subset of replicating genomes

*Wai et al. (2008)* performs BrU labelling to observe the proportion of mitochondria replicating in primary oocytes between P1-4 (21–25 dpc on our time axis). The observations contained therein (*Appendix figure 2* in *Wai et al., 2008*) show a small subset of BrU-labelled mitochondrial foci compared to the overall population of mitochondria labelled with another dye. Here we show that this observation is compatible with (and expected from) our proposed model of random mtDNA turnover.

Consider a population of mtDNAs replicating with rate $\lambda$ and degrading with rate $\nu$. We model the BrU labelling assay as follows. At time $t = 0$, we begin the BrU labelling, which we conservatively model as a perfect process, so that every mtDNA that replicates becomes labelled. We continue this labelling until $t = t^*$, when we observe the proportion of labelled mtDNAs.

For simplicity, we will consider a fixed population of mtDNAs of size $N$, though this reasoning extends to changing population size. We denote by $l$ the number of labelled mtDNAs. After BrU exposure, this number may change in three ways: (A) a replication event involving a previously unlabelled mtDNA will produce two new labelled mtDNAs; (B) a replication event involving a previously labelled mtDNA will produce one new labelled mtDNA; (C) a degradation event involving a labelled mtDNA will remove one labelled mtDNA. The dynamics of labelled mtDNA number during BrU exposure are given by

$$\frac{dl}{dt} = (A) + (B) + (C), \tag{2}$$

$$= 2\lambda(N - l) + \lambda l - \nu l. \tag{3}$$

Assuming that $l = 0$ at $t = 0$, the solution of this equation, for the number of labelled mtDNAs at time $t^*$, is

$$l = \frac{2N\lambda}{\lambda + \nu}\left(1 - e^{-(\lambda+\nu)t^*}\right). \tag{4}$$

Assuming a constant population size requires $\lambda = \nu$. The conclusions of this illustrative study do not substantially change if we allow ($\lambda \neq \nu$) and hence an increasing or decreasing population. We consider the values of $\lambda$ and $\nu$ required to yield values of $l$ comparable with those found in **Wai et al. (2008)**. We will roughly estimate these values, based on the proportion of labelled foci observable, as $l = 0.5\,N$ for 24 hr BrU exposure (half of observed mtDNAs being labelled) and $l = 0.05\,N$ for 2 hr BrU exposure (5% of observed mtDNAs being labelled). A value of $\lambda = \nu = 0.014$ hr$^{-1}$ yields $l = 0.49\,N$ at $t^* = 24$ hr and $l = 0.055\,N$ at $t^* = 2$ hr.

**Figure 3** in our Main text gives the posterior distribution on $\nu$, characterising the rate of random mtDNA turnover in our model, at different times. It can be seen that a value of $\nu = 0.35$ day$^{-1}$ comfortably falls within the region of high posterior density during the time range 21–25 dpc—lying immediately before the strong increase in random turnover that our model subsequently predicts. Our inferred mechanism of random mtDNA turnover is thus compatible with the observations of a labelled subset of mtDNAs in the BrU incorporation assay in **Wai et al. (2008)**—we would expect to see roughly the observed labelling proportion simply due to the likely rates of random mtDNA turnover inferred at that stage of development. Furthermore, we can use this line of reasoning to produce a testable prediction: similar experiments carried out several days later—when random mtDNA turnover is inferred to increase substantially—should show a larger subset of labelled mtDNAs for the same BrU exposure.

## Experimental elucidation

In **Table 3** we list several classes of potential experimental protocols that would assist in further elucidation of the bottlenecking mechanism and our predictions. Potentially useful results include further characterisation of the microscopic detail underlying mtDNA dynamics during development, confirmation of our random turnover model, assessing degree to which heteroplasmy modulates copy number dynamics and exploring our predictions relating mitophagy and bottlenecking power.

**Table 3**. Experiments for further elucidation of the mtDNA bottleneck

| Measurement | Purpose |
|---|---|
| MtDNA copy number before and after cell divisions and/or variance of copy number between daughter cells | To elucidate mechanism of mtDNA partitioning and whether this partitioning is deterministic or stochastic |
| Copy number trajectories with different mtDNA heteroplasmies | To assess the modulation of copy number dynamics by mtDNA heteroplasmy via retrograde signalling |
| Measurement of mean heteroplasmy through development, with a variety of mtDNA type pairings | To assess and quantify to what extent selection modulates mtDNA dynamics during germline development |
| Copy number measurements after upregulation of mitophagy | To assess the presence and strength of compensatory mechanisms that may act to preserve mtDNA copy number—and hence whether upregulating mitophagy will act to increase mtDNA turnover or simply lower copy number |
| Heteroplasmy variance after upregulation of mitophagy | To assess the efficacy of mitophagy for increasing the power of the bottleneck |
| Heteroplasmy distribution in cells after the bottleneck from sampled/known initial heteroplasmy | To confirm predictions for threshold crossing and statistics between generations |
| BrU incorporation in oocytes between 30 and 40 dpc | To confirm the random turnover mechanism: we expect a large proportion of BrU incorporation subset of mtDNAs to be observed in this time period (see section 'Observation of a subset of replicating genomes') |

*Table 3. Continued on next page*

Johnston *et al*. eLife 2015;4:e07464. DOI: 10.7554/eLife.07464

*Table 3. Continued*

| Measurement | Purpose |
|---|---|
| Mitochondrial ultrastructure and mtDNA localisation during development | To assess and characterise any potential modulation of the size of units of mitochondrial inheritance by mitochondrial dynamics through development, in particular, investigating whether there is time-varying modulation of cluster size at points of division |

## Mitophagy regulation

The results from our model suggest a potential clinical pathway for increasing heteroplasmy variance, and thus the power of the bottleneck to remove heteroplasmic cells. We have shown that upregulation of mtDNA degradation (e.g., through increasing mitophagy) leads to lower mtDNA copy numbers and greater heteroplasmy variance. It is unclear whether a given treatment will have the sole effect of upregulating mitophagy: it seems likely that compensatory mechanisms (which we do not explicitly model, but may include retrograde signalling [**Chae et al., 2013**]) will engage to stabilise mtDNA copy number. However, such mechanisms would most straightforwardly be expected to act through increasing mtDNA proliferation, thus having the net effect of increasing mtDNA turnover. We have shown that such an increase in turnover also increases the heteroplasmy variance in a population. We therefore propose that upregulating mitophagy may be a fruitful pathway of investigation for increasing bottlenecking power, either as a standalone effect or due to the action of compensatory mechanisms it may invoke.

Speculatively, potential strategies to upregulate mitophagy may include the limited use of uncouplers to accelerate the mitophagy normally involved in quality control (**de Vries et al., 2012**); targeted chemical treatments with agents that have been identified as regulating mitophagy, including glutathione in yeast (**Deffieu et al., 2009**) and $C_{18}$-pyridium ceramide in human cancer cells (**Sentelle et al., 2012**); modulation of mitochondrial ultrastructure and dynamics to upregulate fission, intrinsically linked to the process of mitophagy (**Youle and Narendra, 2010**; **Twig and Shirihai, 2011**); or the use of existing drugs which have been found to modulate mitophagy, such as Efavirenz (**Apostolova et al., 2011**).

## Heteroplasmy statistics

We have defined heteroplasmy by

$$h = \frac{M_2}{M_1 + M_2}. \tag{5}$$

To find statistics for this quantity we consider the Taylor expansion of a function $f(X_1, X_2)$ of two random variables $X_1, X_2$ about a point $(\mu_1, \mu_2)$, where $\mu_i = \mathbb{E}(X_i)$. We assume that the moments of $X_i$ are well-defined and both have zero probability mass at $X_i = 0$. The Taylor expansion is:

$$f(X_1, X_2) = f(\mu_1, \mu_2) + f_1(\mu_1, \mu_2)(X_1 - \mu_1) + f_2(\mu_1, \mu_2)(X_2 - \mu_2) + \text{higher order terms}, \tag{6}$$

where $f_i$ denotes the derivative of $f$ with respect to $X_i$. We truncate the expansion at first order for later algebraic simplicity, noting that even with this level of precision, the agreement between the resulting analysis and numerical simulation is excellent. Then

$$\mathbb{E}(f(X_1, X_2)) = \mathbb{E}(f(\mu_1, \mu_2) + f_1(\mu_1, \mu_2)(X_1 - \mu_1) + f_2(\mu_1, \mu_2)(X_2 - \mu_2) + \ldots). \tag{7}$$

We note that $\mathbb{E}(X_i - \mu_i) = 0$, so

$$\mathbb{E}(f(X_1, X_2)) \simeq f(\mu_1, \mu_2). \tag{8}$$

Similarly,

$$\mathbb{V}(f(X_1, X_2)) = \mathbb{E}\left(\left(f(X_1, X_2) - \mathbb{E}(f(X_1, X_2))\right)^2\right), \tag{9}$$

$$\simeq \mathbb{E}\left(\left(f(X_1, X_2) - f(\mu_1, \mu_2)\right)^2\right), \tag{10}$$

$$= \mathbb{E}\left(\left(f_1(\mu_1, \mu_2)(X_1 - \mu_1) + f_2(\mu_1, \mu_2)(X_2 - \mu_2)\right)^2\right), \tag{11}$$

and noting that $\mathbb{E}((X_i - \mu_i)^2) = \mathbb{V}(X_i)$ we obtain

$$\mathbb{V}\left(f(X_1, X_2)\right) \simeq \left(f_1(\mu_1, \mu_2)\right)^2 \mathbb{V}(X_1) + \left(f_2(\mu_1, \mu_2)\right)^2 \mathbb{V}(X_2) + 2f_1(\mu_i, \mu_2)f_2(\mu_1, \mu_2)\mathbb{C}(X_1, X_2), \tag{12}$$

where $\mathbb{C}(X_1, X_2)$ is the covariance of $X_1$ and $X_2$. If we now use $f(X_1, X_2) = \frac{X_1}{X_2}$, we have $f_1 = X_2^{-1}$, $f_2 = -X_1 X_2^{-2}$; so

$$\mathbb{E}(X_1/X_2) \simeq \mathbb{E}(X_1)/\mathbb{E}(X_2), \tag{13}$$

$$\mathbb{V}(X_1/X_2) \simeq \mathbb{V}(X_1)/\mathbb{E}(X_2)^2 + \mathbb{E}(X_1)^2 \mathbb{V}(X_2)/\mathbb{E}(X_2)^4 - 2\mathbb{E}(X_1)\mathbb{C}(X_1, X_2)/\mathbb{E}(X_2)^3. \tag{14}$$

If $X_1 = M_2$ and $X_2 = M_1 + M_2$, and $M_1$ and $M_2$ are independent (due to the lack of coupling between the mtDNA species), $\mathbb{C}(X_1, X_2) = \mathbb{V}(M_2)$, and so

$$\mathbb{E}(h) = \frac{\mathbb{E}(M_2)}{\mathbb{E}(M_1) + \mathbb{E}(M_2)}, \tag{15}$$

$$\mathbb{V}(h) = \left(\frac{\mathbb{E}(M_2)}{\mathbb{E}(M_1) + \mathbb{E}(M_2)}\right)^2 \times \left(\frac{\mathbb{V}(M_2)}{\mathbb{E}(M_2)^2} - \frac{2\mathbb{V}(M_2)}{\mathbb{E}(M_2)\left(\mathbb{E}(M_1) + \mathbb{E}(M_2)\right)} + \frac{\mathbb{V}(M_1) + \mathbb{V}(M_2)}{\left(\mathbb{E}(M_1) + \mathbb{E}(M_2)\right)^2}\right). \tag{16}$$

## Derivation of analytic results for binomial model
### Generating function within a cell cycle
To make analytic progress describing the mitochondrial content of quiescent cells, and within a single cell cycle of dividing cells, we use a birth and death model to describe mitochondrial evolution. Without cell divisions, the dynamics of a population of replicating and degrading entities is given by the master equation

$$\frac{dP(m, t)}{dt} = \nu(m+1)P(m+1, t) + \lambda(m-1)P(m-1, t) - (\nu + \lambda)mP(m, t), \tag{17}$$

$$P(m, 0) = \delta_{mm_0}, \tag{18}$$

with $P(m)$ the probability of observing the system with a copy number $m$ at time $t$, and $m_0$ the initial copy number. The corresponding generating function, using the transformation $G(z, t) = \sum_m z^m P(m, t)$, obeys

$$\frac{\partial G(z, t)}{dt} = \left(\nu(1-z) + \lambda(z^2 - z)\right)\frac{\partial G(z, t)}{\partial z}, \tag{19}$$

$$G(z, 0) = z^{m_0}, \tag{20}$$

which is straightforwardly solved by

$$G_0(z, t \mid m_0) = \left(\frac{(z-1)\nu e^{(\lambda - \nu)t} - \lambda z + \nu}{(z-1)\lambda e^{(\lambda - \nu)t} - \lambda z + \nu}\right)^{m_0}, \tag{21}$$

$$\equiv [g(z, t)]^{m_0}, \tag{22}$$

where the 0 subscript signifies that no divisions have occurred, and we have specifically labelled the base of $G_0$ as $g_0$ for later convenience.

## Generating function over cell divisions

We now consider a system undergoing cell divisions. Now, we have a population of organelles with time evolution described by a generating function $G = [g]^{m_0}$ and subject to binomial partitioning at cell division. The probability distribution of $m$ after a single cell division is:

$$P_1(m,t \,|\, m_0) = \sum_{m_{1,b}=0}^{\infty} \sum_{m_{1,a}=0}^{m_{1,b}} P_0(m,t \,|\, m_{1,a}) \binom{m_{1,b}}{m_{1,a}} 2^{-m_{1,b}} P_0(m_{1,b}, \tau \,|\, m_0), \tag{23}$$

where $m_{i,a}$, $m_{i,b}$ mean respectively the number of individuals after and before the $i$th cell division, and the subscript in $P_0$ denotes the fact that this function refers to time evolution within a cell cycle (with no division). The sum takes into account all possible configurations of the system up to the cell division then all possible configurations afterwards, with weighting according to a binomial partitioning. This line of reasoning can straightforwardly be extended to $n$ cell divisions (**Rausenberger and Kollmann, 2008**):

$$P_n(m,t \,|\, m_0) = \sum_{m_{n,b}=0}^{\infty} \sum_{m_{n,a}=0}^{m_{n,b}} \cdots \sum_{m_{1,b}=0}^{\infty} \sum_{m_{1,a}=0}^{m_{1,b}} P_0(m,t \,|\, m_{n,a}) \prod_{i=1}^{n-1} \Phi_i, \tag{24}$$

where $\Phi_i$ is a 'probability propagator' of the form

$$\Phi_i = \binom{m_{i,b}}{m_{i,a}} 2^{-m_{i,b}} P_0(m_{i,b}, \tau \,|\, m_{i+1,a}), \tag{25}$$

and $m_{n+1,a} \equiv m_0$. For clarity, we introduce the nomenclature:

$$\sum_{i,j}{}' \equiv \sum_{m_{i,b}=0}^{\infty} \sum_{m_{i,a}=0}^{m_{i,b}} \cdots \sum_{m_{j,b}=0}^{\infty} \sum_{m_{j,a}=0}^{m_{j,b}}. \tag{26}$$

Now consider the generating function of $P_n$:

$$G_n(z,t \,|\, m_0) = \sum_m z^m P_n(m,t \,|\, m_0), \tag{27}$$

$$= \sum_m \sum_{n,1}{}' z^m P_0(m,t \,|\, m_{n,a}) \prod_{i=1}^{n-1} \Phi_i, \tag{28}$$

$$= \sum_{n,1}{}' G_0(z,t \,|\, m_{n,a}) \prod_{i=1}^{n-1} \Phi_i, \tag{29}$$

$$= \sum_{n-1,1}{}' \sum_{m_{n,b}=0}^{\infty} \underbrace{\sum_{m_{n,a}=0}^{m_{n,b}} [g_0(z,t)]^{m_{n,a}} \binom{m_{n,b}}{m_{n,a}} 2^{-m_{n,b}}}_{\text{binomial term}} P_0(m_{n,b}, \tau \,|\, m_{n-1,a}) \prod_{i=1}^{n-2} \Phi_i, \tag{30}$$

$$= \sum_{n-1,1}{}' \underbrace{\sum_{m_{n,b}=0}^{\infty} \left( \frac{1}{2} + \frac{g_0(z,t)}{2} \right)^{m_{n,b}}}_{\text{generating function with transformed variable}} P_0(m_{n,b}, \tau \,|\, m_{n-1,a}) \prod_{i=1}^{n-1} \Phi_i, \tag{31}$$

$$\equiv \sum_{n-1,1}{}' \overbrace{G_0(z',\tau \,|\, m_{n-1,a})} \prod_{i=1}^{n-2} \Phi_i, \tag{32}$$

where we have used the identity $\sum_{a=0}^{b} x^a \binom{b}{a} 2^{-b} \equiv \left( \frac{1}{2} + \frac{x}{2} \right)^b$ and changed variables $z' = \frac{1}{2} + \frac{g_0(z,t)}{2}$. Comparing **Equations 29, 32** and following this process by induction we can see that the overall generating function is $G_n = h_0^{m_0}$, where $h$ is the solution to the recursive system

$$h_i = g_0\left( \frac{1}{2} + \frac{h_{i+1}}{2}, \tau \right), \tag{33}$$

$$h_n = g_0(z, t). \tag{34}$$

$h_i$ is of the form $\frac{ah_{i+1} + b}{ch_{i+1} + d}$ (from **Equation 21**). This expression takes the form of a Riccati difference equation and can be solved exactly after **Brand (1955)**. The solution is straightforward but algebraically lengthy, and we defer presentation of the full procedure to a future technical publication (**Johnston and Jones, 2015**). The overall solution is:

$$G_C(z, t, n) = h_0 = \frac{2^n(I-2)(\lambda z - \nu) + I'(z-1)\left(\left(\lambda(2^n - I^n) - \nu I^n(I-2)\right)\right)}{\lambda I'(z-1)(2^n + I^n - I^{n+1}) + 2^n(I-2)(\lambda z - \nu)}, \tag{35}$$

where the $C$ subscript denotes cycling cells, and

$$I = e^{(\lambda - \nu)\tau}, \tag{36}$$

$$I' = e^{(\lambda - \nu)t}. \tag{37}$$

## Generating function for different phases

We now consider how to extend this reasoning to the overall bottlenecking process, which in general may involve several phases of quiescent and cycling dynamics with different kinetic parameters. We begin with the generating function bases $g_i(z, t)$ for each regime $i$. For consistency with the above approach, we label phases starting from a zero index, so the first phase corresponds to $i = 0$, and we use $i_{max}$ to denote the label of the final phase. Then we use

$$h_{i_{max}} = g_i(z, t), \tag{38}$$

$$h_i = g_i(h_{i+1}, 0), \tag{39}$$

$$G_{overall} = h_0^{m_0}, \tag{40}$$

using induction over the different phases in the way we used induction over different cell cycles above. Here we consider the changeover between regimes by using the generating function at the start of the incoming phase.

The appropriate generating function bases for quiescent (**Equation 21**) and cycling (**Equation 35**) cells can be written as

$$g_Q(z, t \mid m_0) = \left(\frac{A_Q z + B_Q}{C_Q z + D_Q}\right), \tag{41}$$

$$g_C(z, t, n \mid m_0) = \left(\frac{A_C z + B_C}{C_C z + D_C}\right), \tag{42}$$

with coefficients

$$A_Q = \nu I' - \lambda, \tag{43}$$

$$B_Q = \nu - \nu I', \tag{44}$$

$$C_Q = \lambda I' - I', \tag{45}$$

$$D_Q = \nu - \lambda I', \tag{46}$$

$$A_C = 2^n \lambda(I + I' - 2) - I^n I'\left(\lambda + \nu(I - 2)\right), \tag{47}$$

$$B_C = I^n I'\left(\lambda + \nu(I - 2)\right) - 2^n\left(\lambda I' + \nu(I - 2)\right), \tag{48}$$

$$C_C = -\lambda I^n I'(I - 1) + 2^n \lambda(I + I' - 2), \tag{49}$$

$$D_C = \lambda l^n l'(l-1) - 2^n(\lambda l' + \nu(l-2)), \tag{50}$$

using, as before, $l = e^{(\lambda - \nu)\tau}$ and $l' = e^{(\lambda - \nu)t}$. Note that the cycling coefficients reduce to the quiescent coefficients when $n \to 0$ and $\tau \to 0$. The values of the appropriate $A$, $B$, $C$, $D$ coefficients for a given dynamic phase thus follow straightforwardly from the kinetic parameters of that phase, with the appropriate choice between quiescent and cycling parameters being made.

If we now label these coefficients with a subscript denoting the appropriate phase of bottlenecking, so that, for example, $A_i$ is **Equation 47** with $\lambda_i$, $\nu_i$, $n_i$ replacing $\lambda$, $\nu$, $n$, we can write:

$$h_{i_{max}} = \frac{A_{i_{max}} z + B_{i_{max}}}{C_{i_{max}} z + D_{i_{max}}}, \tag{51}$$

$$h_i = \frac{A_i h_{i+1} + B_i}{C_i h_{i+1} + D_i}, \tag{52}$$

$$g_{overall} = h_0. \tag{53}$$

Following this recursion for $n$ phases of bottlenecking and simplifying the resultant multi-layer fraction gives rise to the solution

$$g_{overall} = h_0 = \frac{A'z + B'}{C'z + D'}, \tag{54}$$

where

$$\begin{bmatrix} A' & B' \\ C' & D' \end{bmatrix} = \prod_{i=1}^{n} \begin{bmatrix} A_i & B_i \\ C_i & D_i \end{bmatrix}, \tag{55}$$

from which $G_{overall} = g_{overall}^{m_0}$ follows straightforwardly. The following results will be of assistance:

$$\mathbb{E}(m) = \frac{d}{dz}\left(\frac{A'z + B'}{C'z + D'}\right)^{m_0}\bigg|_{z=1} = \frac{m_0(A'D' - B'C')\left(\frac{A'+B'}{C'+D'}\right)^{m_0-1}}{(C'+D')^2}, \tag{56}$$

$$\frac{d^2}{dz^2}\left(\frac{A'z + B'}{C'z + D'}\right)^{m_0}\bigg|_{z=1} = \frac{m_0(B'C' - A'D')\left(\frac{A'+B'}{C'+D'}\right)^{m_0}\left(B'C'(m_0+1) + A'(2C' + D'(1-m_0))\right)}{(A'+B')^2(C'+D')^2}, \tag{57}$$

$$= -\frac{\left(\frac{A'+B'}{C'+D'}\right)}{(A'+B')^2}\left(B'C'(m_0+1) + A'(2C' + D'(1-m_0))\right)\mathbb{E}(m). \tag{58}$$

As **Equations 43–46** can be thought of as special cases of **Equations 47–50**, we combine **Equations 47–50** into **Equation 55**, and, simplifying, we find the following relations:

$$A' + B' = C' + D' = \prod_{phase\ i} 2^{n_i}(l_i - 2)(\lambda_i - \nu_i), \tag{59}$$

$$A'D' - B'C' = \prod_{phase\ i} 2^{n_i}(l_i - 2)^2 l_i^{n_i} l'_i(\lambda_i - \nu_i)^2; \tag{60}$$

we then immediately obtain

$$\mathbb{E}(m) = m_0 \prod_i \left( \frac{2^{n_i}(l_i-2)^2 l_i^{n_i} l'_i (\lambda_i - \nu_i)^2}{4^{n_i}(l_i-2)^2(\lambda_i - \nu_i)^2} \right), \tag{61}$$

$$= m_0 \prod_i 2^{-n_i} l_i^{n_i} l'_i, \tag{62}$$

$$\mathbb{V}(m) = \frac{-\left(B'C'(m_0+1) + A'\left(2C' + D'(1-m_0)\right)\right)}{\prod_i 4^{n_i}(l_i-2)^2(\lambda_i - \nu_i)^2} \mathbb{E}(m) + \mathbb{E}(m) - \mathbb{E}(m)^2, \tag{63}$$

leaving us only with the problem of calculating the expression $(B'C'(m_0+1) + A'(2C' + D'(1-m_0)))$ in the variance calculation. We were not able to dramatically simplify this expression and so, for clarity, write:

$$\Phi = -\left(B'C'(m_0+1) + A'\left(2C' + D'(1-m_0)\right)\right), \tag{64}$$

which gives us:

$$\mathbb{V}(m) = \frac{\Phi \mathbb{E}(m)}{\prod_i 4^{n_i}(l_i-2)^2(\lambda_i - \nu_i)^2} + \mathbb{E}(m) - \mathbb{E}(m)^2. \tag{65}$$

We note that $\Phi$ is just a notational simplification and is straightforwardly calculable by inserting *Equations 47–50* into *Equation 55* then computing *Equation 64*.

## Constant population size

For generality, we consider enforcing a constant population size in post-mitotic cells (not undergoing divisions). This process involves setting $\lambda = \nu$, so the net gain in mtDNA is zero. If we write $\lambda = \nu + \epsilon$ and take the limit $\epsilon \to 0$, *Equation 21* becomes

$$G_{c,post}(z,t) = \left( \frac{\nu t z - z - \nu t}{\nu t z - 1 - \nu t} \right)^{m_0}. \tag{66}$$

To enforce a constant mean population size in mitotic cells, it is necessary to balance the expected loss of mtDNA through repeated divisions with an expected increase during the cell cycle. This balance can be accomplished by setting $\lambda = \nu + \frac{\ln 2}{\tau}$. Writing $\lambda = \nu + \frac{\ln 2}{\tau} + \epsilon$ and taking the $\epsilon \to 0$ limit we obtain

$$G_{c,mito}(z,t) = \left( \frac{2\nu\tau(z-1) - 2^{t/\tau}(z-1)\left((n_1+2)\nu\tau + n_1 \ln 2\right) + z\ln 4}{2\nu\tau(z-1) - 2^{t/\tau}(z-1)(n_1+2)(\nu\tau + \ln 2) + z\ln 4} \right)^{m_0}. \tag{67}$$

In both these cases, the same approach as above can be used to derive moments of the resulting probability distributions.

## Explicit distributions

The probability of observing exactly $m$ mtDNAs of a given type can be found from the generating function with

$$\mathbb{P}(m,t) = \frac{1}{m!} \frac{\partial^m}{\partial z^m} G(z,t) \bigg|_{z=0}. \tag{68}$$

We can use Leibniz' rule on a generating function of form $G = \left( \frac{A'z + B'}{C'z + D'} \right)^{m_0}$ by setting $f \equiv (A'z + B')^{m_0}$, $g \equiv (C'z + D')^{m_0}$ and writing

$$\frac{\partial^m G}{\partial z^m} = \sum_{k=0}^{m} \binom{m}{k} \frac{\partial^k f}{\partial z^k} \frac{\partial^{(m-k)} g}{\partial z^{(m-k)}}, \tag{69}$$

$$= \sum_{k=0}^{m} \binom{m}{k} (A')^k (A'z + B')^{m_0-k} \frac{m_0!}{(m_0-k)!} (C')^{m-k} (C'z + D')^{(-m_0-m-k)} (-1)^{m-k} \frac{(m_0+m-k-1)!}{(m_0-1)!}.$$

(70)

Enforcing $z = 0$ and rewriting in terms of a hypergeometric function gives

$$\mathbb{P}(m,t) = \frac{1}{m!} \frac{(-1)^m (B')^{m_0} (C')^m (D')^{-m-m_0} (m_0+m-1)!}{(m_0-1)!} {}_2F_1 \left(-m, -m_0; 1-m-m_0; \frac{A'D'}{B'C'}\right).$$ (71)

The distribution of heteroplasmy is then given by

$$\mathbb{P}(h) = \sum_{m_1=0}^{\infty} \sum_{m_2=0}^{\infty} \mathbb{P}(m_1, t \mid (1-h_0)m_0) \mathbb{P}(m_2, t \mid h_0 m_0) \mathscr{I} \left(\frac{m_2}{m_1+m_2}, h\right),$$

(72)

where $\mathscr{I}(h', h)$ is an indicator function returning 1 if $h' = h$ and 0 otherwise. Computing the probability of observing a given heteroplasmy thus involves a sum, over all mtDNA states that correspond to that heteroplasmy, of the probability of that state.

The evaluation of hypergeometric functions is more computationally demanding than that of more common mathematical functions, and the infinite sums at first glance seem intractable. However, in practise and using parameterisations from our inferential approach, vanishingly little probability density exists at $m_1, m_2 > 5 \times 10^5$, corresponding to the biological observation that mtDNA copy number is very unlikely to exceed this value. Dynamic programming then allows these sums to be performed straightforwardly.

Finally, the computation of $\mathbb{P}(m=0, t)$ is important in our analysis of the characterisation of key distributions using the first two moments (see below), where it appears as $\mathbb{P}(m_2 = 0, t)$, the probability of wildtype fixation. This is relatively straightforward to address analytically as when $m = 0$, **Equation 68** reduces to $\mathbb{P}(0, t) = |G_{overall}|_{z=0}$, which in the notation above is simply:

$$\zeta \equiv \mathbb{P}(0, t) = \left(\frac{B'}{D'}\right)^{m_0},$$

(73)

where we introduce the notation $\zeta$ for fixation probability for later brevity. We could not dramatically simplify the full expression so we leave it in this form and note that it can be readily calculated (as above) by inserting **Equations 47–50** into **Equation 55** then computing **Equation 73**.

## Multiple species and heteroplasmy

The heteroplasmy $h = m_2/(m_1 + m_2)$ is straightforwardly addressable by considering the above solutions for $m_1$ and $m_2$. We can also consider a more general case, in which we have four species of mtDNA in our model: wildtype reproducing ($m_1$), mutant reproducing ($m_2$), wildtype sterile ($m_3$) and mutant sterile ($m_4$). We assume that these species evolve in an uncoupled way with time. The parameter $h_0$, initial heteroplasmy, determines the initial proportion of mutant genomes: $h_0 = \frac{m_{20} + m_{40}}{m_0}$, where $m_0 = m_{10} + m_{20} + m_{30} + m_{40}$ is the total initial copy number of mtDNA. The parameter $\alpha$ determines the proportion of genomes capable of reproducing: $\alpha = \frac{m_{10}}{m_{10} + m_{30}} = \frac{m_{20}}{m_{20} + m_{40}}$. We compute the time trajectories for all $m_i$ then calculate heteroplasmy by setting $M_1 = m_1 + m_3$, $M_2 = m_2 + m_4$, respectively the total numbers of wildtype and mutant mtDNAs, and using **Equations 15, 16**, where all means and variances are straightforwardly extracted from the above analysis.

## Characterisation of distributions of important quantities with moments

We are interested in the probability with which heteroplasmy $h$ exceeds a certain threshold value $h^\star$. This probability can be computed using **Equation 72** above, but the large sums of

hypergeometric functions suggest that a simpler approximation of the heteroplasmy distribution may be desirable, both for computational simplicity and intuitive interpretability. We here explore how well distributions of copy number and, importantly, heteroplasmy are characterised by quantities that are easily obtained from our analytic approaches without large summations: specifically, low-order moments $\mathbb{E}(m)$, $\mathbb{V}(m)$, and fixation probabilities $\mathbb{P}(m=0)$.

For moderate initial heteroplasmy $0.7 > h_0 > 0.3$, all distributions are well matched by the Normal distributions computed using the first two moments $\mathbb{E}(m)$ and $\mathbb{V}(m)$. This match begins to fail as initial heteroplasmy decreases or increases to the extent where fixation of one mtDNA type becomes likely. The resultant non-negligible probability density at $h = 0$ and/or $h = 1$ represents a truncation point which forces skew on the distributions (particularly $\mathbb{P}(h)$) and weakens the Normal approximation.

We can make progress by considering $\mathbb{P}(h)$ to be a weighted sum of a truncated Normal distribution $\mathcal{N}'(\mu, \sigma^2)$ (truncated at 0, 1; and with currently unknown parameters $\mu$, $\sigma$) and two $\delta$-functions at $h = 0$ and $h = 1$ representing the fixation probability of wildtype and mutant mtDNA respectively. If we write $\mathbb{P}_{\mathcal{N}'}(h)$ for the probability density at $h$ of such a truncated Normal distribution, we have:

$$\mathbb{P}(h) = (1 - \zeta_1 - \zeta_2)\mathbb{P}_{\mathcal{N}'}(h) + \zeta_1\delta(h) + \zeta_2\delta(h-1), \tag{74}$$

where $\zeta_1 = \mathbb{P}(m_2 = 0, t)$ is the fixation probability of the wildtype and $\zeta_2 = \mathbb{P}(m_1 = 0, t)$ is the fixation probability of the mutant, expressions for which were computed previously in **Equation 73**. Knowledge of the parameters $\mu$, $\sigma$ that describe the truncated Normal part of this distribution will then provide us with a better estimate of $\mathbb{P}(h)$.

We can use the relations $\mathbb{E}(h) = \int h\mathbb{P}(h)dh$ and $\mathbb{V}(h) = \int h^2\mathbb{P}(h)dh - \mathbb{E}(h)^2$. As $\delta(h)$ provides a nonzero contribution to these integrals only when $h = 0$, the contribution from this part of $\mathbb{P}(h)$ is always zero; then,

$$\mathbb{E}(h) = \int h\big((1 - \zeta_1 - \zeta_2)\mathbb{P}_{\mathcal{N}'}(h) + \zeta_2\delta(h-1)\big)dh, \tag{75}$$

$$= (1 - \zeta_1 - \zeta_2)\mathbb{E}(\mathcal{N}') + \zeta_2, \tag{76}$$

$$\mathbb{V}(h) = \int h^2\big((1 - \zeta_1 - \zeta_2)\mathbb{P}_{\mathcal{N}'}(h) + \zeta_2\delta(h-1)\big)dh - \mathbb{E}(h)^2, \tag{77}$$

$$= (1 - \zeta_1 - \zeta_2)\big(\mathbb{V}(\mathcal{N}'_{h>0}) + \mathbb{E}(\mathcal{N}'_{h>0})^2\big) - \mathbb{E}(h)^2 + \zeta_2, \tag{78}$$

where $\mathbb{E}(\mathcal{N}')$, $\mathbb{V}(\mathcal{N}')$ are respectively the mean and variance of the truncated Normal distribution, and in the final line we have used the fact that $\mathbb{V}(\mathcal{N}') = \int h^2\mathbb{P}_{\mathcal{N}'}(h)dh - \mathbb{E}(\mathcal{N}')^2$.

Results are known (**Greene, 2003**) for moments of the truncated Normal distribution:

$$\mathbb{E}(\mathcal{N}') = \mu + \sigma\frac{f(\alpha_1) - f(\alpha_2)}{F(\alpha_2) - F(\alpha_1)}, \tag{79}$$

$$\mathbb{V}(\mathcal{N}') = \sigma^2\left(1 - \frac{\alpha_1 f(\alpha_1) - \alpha_2 f(\alpha_2)}{F(\alpha_2) - F(\alpha_1)} - \left(\frac{f(\alpha_1) - f(\alpha_2)}{F(\alpha_2) - F(\alpha_1)}\right)^2\right), \tag{80}$$

where, in our case (with truncations at $h = 0$ and $h = 1$) $\alpha_1 = -\mu/\sigma$, $\alpha_2 = (1 - \mu)/\sigma$ and $f(x) = (\sqrt{2\pi})^{-1}\exp(-x^2/2)$ and $F(x) = \frac{1}{2}(1 + \mathrm{erf}(x/\sqrt{2}))$ are respectively the p.d.f. and c.d.f. of the standard Normal distribution. Given these expressions, we wish to invert these **Equations 76, 78** to find $\mu$ and $\sigma$, the parameters underlying the truncated Normal distribution, given $\mathbb{E}(h)$, $\mathbb{V}(h)$ and $\zeta_{1,2} = \mathbb{P}(m_{2,1} = 0)$, which we can compute (see below). We have not been able to find an analytic solution for these equations; however, numerically solving these equations is computationally far cheaper than performing the numeric simulations required to better

characterise the real distribution. We then obtain an expression for $\mathbb{P}(h)$, which well matches the exact distribution derived using *Equation 72* (see *Appendix figure 4*).

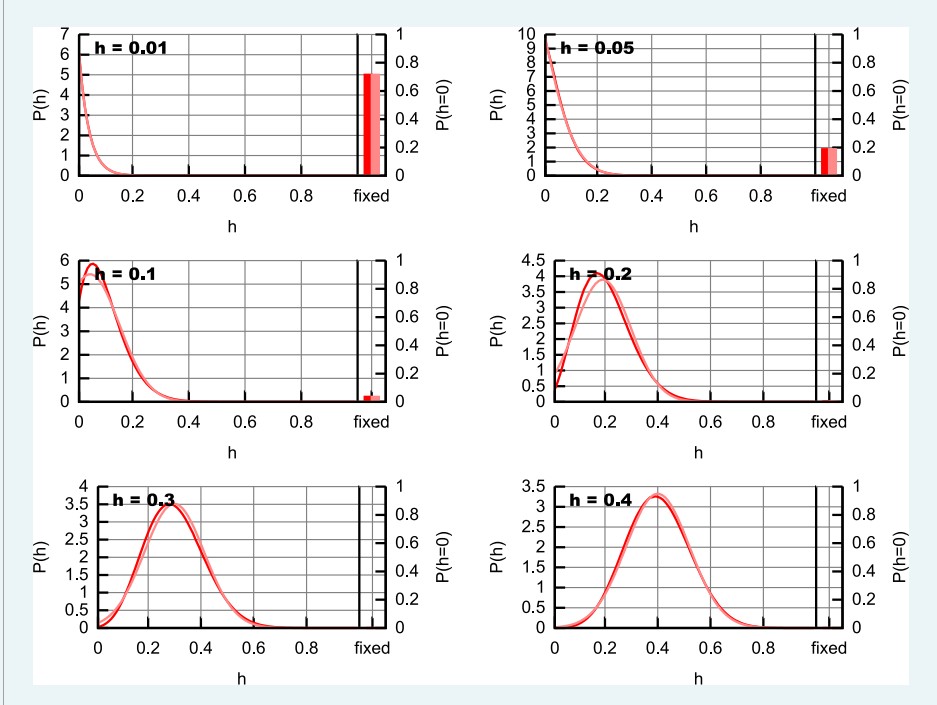

**Appendix figure 4**. Comparison of truncated Normal approximation with exact heteroplasmy distribution. Representations of heteroplasmy distributions at a time $t = 21$ dpc, with various starting heteroplasmies, using (as an example) the maximum likelihood parameterisation emerging from the inference procedure in the Main text. Dark lines and bars show exact distributions from *Equation 72*; pale lines and bars show distributions arising from the truncated Normal distribution described in the text.

## Threshold crossing

The probability of crossing a threshold heteroplasmy $h^*$ with time is simply given by the probability density in the region $h > h^*$. We can then use the result

$$\mathbb{P}(h > h^*) = (1 - \zeta_1 - \zeta_2)\left(1 - \frac{1}{2}\left(1 + \mathrm{erf}\left((h^* - \mu)\big/\sqrt{2\sigma^2}\right)\right)\right) + \zeta_1(1 - \delta(h^*)) + \zeta_2(1 - \delta(h^* - 1)),$$

(81)

for threshold crossing, which follows straightforwardly from considering the integrated density of the model distribution (*Equation 74*) of $h$ above $h^*$, with parts from the error function representing the definite integral of the truncated Normal part of the distribution, with additional terms from wildtype fixation (if $h^* \neq 0$) and mutant fixation (if $h^* \neq 1$).

## Inferring embryonic heteroplasmy

The probability that a sample measurement $h_m$ came from an embryo with heteroplasmy $h_0$ can be found from Bayes' Theorem:

$$\mathbb{P}(h_0 \mid h_m) = \frac{\mathbb{P}(h_m \mid h_0)\mathbb{P}(h_0)}{\mathbb{P}(h_m)}.$$

(82)

We assume a uniform prior distribution $\mathbb{P}(h_0) = \rho$ on embryonic heteroplasmy (though this can be straightforwardly generalised). $\mathbb{P}(h_m)$ is given by the integral over all possible embryonic heteroplasmies of making observation $h_m$, so we obtain

$$\mathbb{P}(h_0 \mid h_m) = \frac{\rho \mathbb{P}(h_m \mid h_0)}{\int_0^1 dh_0' \rho \mathbb{P}(h_m \mid h_0') dh_0'}, \tag{83}$$

$$= \frac{(1 - \zeta_1 - \zeta_2)\mathcal{N}'(h_m \mid \mu, \sigma^2) + \zeta_1 \delta(h_m) + \zeta_2 \delta(h_m - 1)}{\int_0^1 dh_0' \left((1 - \zeta_1 - \zeta_2)\mathcal{N}'(h_m \mid \mu, \sigma^2) + \zeta_1 \delta(h_m) + \zeta_2 \delta(h_m - 1)\right)}, \tag{84}$$

where the $\mu$, $\sigma^2$ moments characterising the truncated Normal distribution are found numerically as above (for each $h_0'$ value in the integrand, which is performed numerically); and $\zeta_1$, $\zeta_2$ are also functions of $h_0$.

