## [Decision Letter]

Thank you for resubmitting your work entitled “Stochastic modelling, Bayesian inference, and new in vivo measurements elucidate the debated mtDNA bottleneck mechanism” for further consideration at *eLife*. Your revised article has been favorably evaluated by Aviv Regev (Senior editor) and a member of the Board of Reviewing Editors along with two reviewers. The manuscript has been improved but there are some remaining issues that need to be addressed before acceptance as detailed in the reviewers' comments below.

In particular, as pointed out in the previous version, claims of novelty in several areas are overstated and require the appropriate revision for accuracy. The reviewers' concerns about data variance in the experimental data also need to be addressed (especially in the 25 day old data) as well as the concern that there is a very large effect with only small alterations in mtDNA degradation.

*Reviewer #1*:

This is a resubmission of a pure modeling paper, with the addition of new experimental data which is presented as supporting the main conclusions of the modeling. The authors have carried out a very thorough theoretical and analytical analysis of the proposed mtDNA genetic bottleneck, and conclude that the existing stance taken by Cree and Wai model (b) is a sensible interpretation of the experimental data. As such, the current paper is reassuringly confirmatory. However, I still have concerns, in part relating to the way the paper is written, and in part relating to the new data, which raises additional questions.

Introduction. A statement is made: ‘No study yet exists combining an analytic ‘bottom-up’ physical description of the bottleneck with mtDNAs as individual, discrete elements subject to a possible variety of replication and partitioning dynamics throughout an explicitly modelled series of cell divisions’. This is not correct, and I made this point in my earlier review of the paper. Their reference (18) includes such a ‘bottom up’ model. Although the compartments did not vary in that model, assumptions were made about the partitioning which are in accordance with the conclusions of this current paper.

Likewise, the statement ‘theory describing the behaviour of mtDNA populations throughout development, as opposed to estimating an overall effect of the bottleneck, is also absent’ is also an over-statement designed to justify the current work. There are several such theories that the authors refer to in their Introduction.

For the new experimental data, they use a line previously reported in their Cell Reports publication. A key message of this publication was that different mtDNA haplotypes behave differently, that selection occurs, and that this is influenced by the nuclear genetic background. All of these effects will influence the heteroplasmy variance. Although the authors touch on selection, they do not explore this thoroughly in the context of their previous work. How will this impact on their interpretation of their bottleneck models?

For the new experimental data, the heteroplasmy measurements are reported to a remarkable degree of accuracy. Do the authors really think that they can reliably report down to 0.01% as indicated in the supplementary data?

Previous theoretical work has shown that ∼50 heteroplasmy measurements are required to reliably compute heteroplasmy variance. Several of the published datasets do not achieve this, and thus the reported variance measurements are likely to be inaccurate. The same may also apply to the new experimental dataset.

Figure 4 raises concerns. For the ∼25 day old data, 6 data points (from a total of 15) are way outside the confidence intervals predicted by the various models. This implies that the new data set is not supporting the current or previous models.

Finally, given the likely readership, it is important to keep reminding readers what is prediction, and what is actual measurement. The section on turnover illustrates this nicely. No turnover measurements have been made; indeed, they would be incredibly difficult to do.

*Reviewer #2*:

The authors have revised their previous submission to now include new experimental data on the mtDNA bottleneck, as requested in an earlier round of review.

There is clearly a great deal of variation in the new data for the normalized variance, but the new data does fit best with the authors' favoured birth-death-partition model. The inclusion of this extra data, at the time points which are most discriminatory with regard to the various models, is therefore helpful, though not absolutely definitive. In particular, it looks as if the spread of the normalised variance itself has interesting dynamics, starting out low, increasing to a maximum at around 25 dpc before diminishing again. Is there any way in which the BDP model can capture these dynamics? Also it was not clear to me why the BDP curve in Figure 4 differs from the HB theory mean plot in Figure 4.

Finally, I remain concerned that a 2% alteration in mtDNA degradation can have such a large effect, see Figure 5. The authors must address this issue.

[Editors’ note: a previous version of this study was rejected after peer review, but the authors submitted for reconsideration. The previous decision letter after peer review is shown below.]

Thank you for choosing to send your work entitled “Stochastic modelling and Bayesian inference elucidate the debated mechanism of the mtDNA bottleneck” for consideration at *eLife*. Your full submission has been evaluated by Aviv Regev (Senior editor), a Reviewing editor and three peer reviewers, and the decision was reached after discussions between the reviewers. We regret to inform you that your work will not be considered further for publication.

Although there was enthusiasm for the quantitative approach to the mitochondrial DNA bottleneck, the reviewers felt that experiments directed at testing the predictions of the modeling are required for the manuscript to significantly advance the field. Therefore, at this stage, we are returning the manuscript to you with an encouragement to submit with experimental data. We hope the reviewers' comments are helpful.

*Reviewer #1*:

The manuscript from Johnston et al. is an extremely thorough and well-thought out attempt to inject rigorous, quantitative thinking into the mechanism of the mtDNA bottleneck. It would appear that prior attempts in this direction have been limited in scope and inconclusive. The strength of the authors' approach is that it is able to unify previous treatments of the phenomenon into a single theoretical framework. From this position, it is possible to assess the plausibility of the various mechanisms within a Bayesian statistical analysis. Using prior data, the authors conclude that a birth-death-partition model is most probable. This model is analytically tractable which is also advantageous. These are powerful conclusions, and it would seem absolutely certain that progress in this field is going to require an analysis of this sort, due to the inherent complexity (and stochasticity) of the problem.

I do however have some concerns. Principally, the analysis is based entirely on previously published data, which is not all internally consistent. In particular, looking at Figure 2 for the variances, there doesn't seem to be very good agreement between the different datasets (especially comparing the Jenuth data with the rest). I appreciate that the authors' statistical analysis takes into account the experimental uncertainty, but the divergence between the different experimental data sets found in the different studies suggests that the effective experimental error bars may be much bigger than estimated. The ability to distinguish between the models via Bayesian model comparison may then be diminished. This point is important, as there is relatively little difference between the models with respect to the mean values (whose experimental values are also well clustered between the various data sets). Hence it does seem that the variances are the key to distinguishing between the models. Also the statistical inference is based around a residual sum of squares difference. However, the way this was implemented (using logs of the copy number measurements, for example) seemed a bit arbitrary. Were other possibilities investigated to ensure consistency?

More generally, I am concerned that the modelling is used primarily as a tool to fit existing data and thereby determine which model is most likely on the basis of previously published data. It would be more powerful if the authors could actually test the novel predictions of their modelling in new experiments. This would elevate the study to a higher level. Their model most definitely is predictive (Table 2), it's just that they ought to consider teaming up with an experimental group and prove these predictions right!

One other issue concerns the sensitivity of the system to variations in the mtDNA degradation rate, where even a 2% change causes a huge difference in the mean and variance (Figure 4), which is probably unlikely in vivo. This may hint at an underlying weakness of the model.

*Reviewer #2*:

This paper deals with a controversy in the mitochondrial inheritance field over the details of the inheritance bottleneck and the development of variability in mtDNA heteroplasmy levels across the cells of an organism. This new mathematical modeling approach is coming from a set of researchers, including one well-known mitochondrial DNA expert, who were not involved in the original set of competing papers (primarily references [18], [67], [72], [57], [12], [13]). These authors also use simulation and mathematical methods that are far more sophisticated than the methods used in previous papers (references [18] and Chinnery et al., 1999).

The paper does a good job of summing up the results and conflicting claims of the earlier literature. Their presentation of the differences in the assumptions of the competing models seems accurate to me, and they have built simulation models that encompass the three general competing sets of assumptions. The experimental data from the previous literature is all presented in a combined manner, which is valuable in itself. Through the Beyesian methods of assessing goodness of fit of the combined set of experimental data to the three competing models, the authors present a clear conclusion concerning the model that has the best agreement with the experimental data. This is a valuable contribution to the literature, and again it is very useful that this assessment is coming from an independent set of researchers.

While the mathematical detail that comes along with these more sophisticated methods is hard to digest, even for a trained mathematician, I would argue that it is essential to have this level of detail in the publication record.

*Reviewer #3*:

The authors have used stochastic modelling and Bayesian inference on published data sets to explore different models of the mitochondrial DNA genetics bottleneck.

There are several statements in the manuscript that are inaccurate and misleading. The notion that organisms ameliorate somatic mutation burden through bottlenecking is a hypothesis (not fact, as written), and not all oocytes have heteroplasmy as stated. Also, the idea that no model has been presented from the ‘bottom up’, with mtDNA as discrete elements, is not correct. The authors cite examples where this was the case (i.e. where the segregating units in the model are proposed to be mtDNA molecules). I also do not accept that there are no methods to allow the comparison of different bottlenecks. One of the authors (Poulton) is a co-author on a Bayesian approach, and Samuels has published extensively on this (some of the papers cited in the current manuscript).

Some of the proposed mechanisms (which are modelled) have little or no experimental basis. For example, we do not know there is ‘turnover’ or mtDNA within germ cells, and certainly the rate is not known. We have no idea what the size of the nucleoid is in germ cells. There is no evidence that turnover is generally low during cell division to my knowledge, to name but a few. The problem is, by including all of these parameters, the potential outcomes of the modelling will increase.

Most importantly, their general conclusions are in keeping with the previous models developed by Samuels, which combine relaxed replication and random segregation contribute to the generation of heteroplasmy in the germ line (i.e. the Cree mechanism, as they describe it). In other words, their findings are confirmatory, not novel.

The theoretical manipulation of the bottleneck provides some interesting results, but these are somewhat obvious: if you reduce mtDNA content (by increasing degragadation; incidentally this is something not shown in the germ line), you will accelerate segregation and thus lead to the loss of mutations that reach high heteroplasmy levels. Likewise, it is not surprising that selective pressures influence heteroplasmy variance. Given that heteroplasmy levels are bound by 0% and 100%, any factors pushing the level in a particular direction will inevitably alter the range of values. It is also not surprising that the observed heteroplasmy variance is consistent with a range of bottleneck sizes, given the number of other parameters that can be varied in the described models.

---

## [Author Response]

Reviewer #1:

*This is a resubmission of a pure modeling paper, with the addition of new experimental data which is presented as supporting the main conclusions of the modeling. The authors have carried out a very thorough theoretical and analytical analysis of the proposed mtDNA genetic bottleneck, and conclude that the existing stance taken by Cree and Wai model (b) is a sensible interpretation of the experimental data. As such, the current paper is reassuringly confirmatory. However, I still have concerns, in part relating to the way the paper is written, and in part relating to the new data, which raises additional questions*.

*Introduction. A statement is made: ‘No study yet exists combining an analytic ‘bottom-up’ physical description of the bottleneck with mtDNAs as individual, discrete elements subject to a possible variety of replication and partitioning dynamics throughout an explicitly modelled series of cell divisions’. This is not correct, and I made this point in my earlier review of the paper. Their reference (*[18]*) includes such a ‘bottom up’ model. Although the compartments did not vary in that model, assumptions were made about the partitioning which are in accordance with the conclusions of this current paper*.

*Likewise, the statement ‘theory describing the behaviour of mtDNA populations throughout development, as opposed to estimating an overall effect of the bottleneck, is also absent’ is also an over-statement designed to justify the current work. There are several such theories that the authors refer to in their Introduction*.

We appreciate the reviewer's guidance on the correct positioning of our paper. We do think that our work is a substantial and original contribution and agree that we must hit on the best way of conveying that contribution without inappropriately diminishing the excellent work of colleagues (we definitely do not want to do that!). The first statement in question contains several important qualifications which are not explicitly noted by the reviewer and which we suggest make our original wording acceptable:

“No study yet exists *combining* an *analytic* ‘bottom-up’ physical description of the bottleneck with mtDNAs as individual, discrete elements *subject to a possible variety* of replication and partitioning dynamics throughout an explicitly modelled series of cell divisions” [our emphasis].

We would contend that this claim is in fact true. We are not claiming that ours is the first bottom-up model of mtDNA dynamics; the novelty we claim is in the combination of our analytic progress (including features like explicit results for heteroplasmy variance with time) with the variety of possible replication and partitioning dynamics that we include. The simulations in [18] do not provide analytic results nor consider other models than relaxed replication with binomial partitioning; previous analytic work has focused on particular specific models (for example, relaxed replication). We therefore contend that the novel combination of analysis and generality we claim are indeed true. We have reworded the text concerned to clarify that the novelty we claim lies in the variety of situations we consider (and thus the ability to select between these situations) and in the analytic progress we make.

We accept that our second sentence, regarding previous work modelling mtDNA through development, was still ambiguously phrased. We have simplified and just removed this sentence from the Introduction.

*For the new experimental data, they use a line previously reported in their Cell Reports publication. A key message of this publication was that different mtDNA haplotypes behave differently, that selection occurs, and that this is influenced by the nuclear genetic background. All of these effects will influence the heteroplasmy variance. Although the authors touch on selection, they do not explore this thoroughly in the context of their previous work. How will this impact on their interpretation of their bottleneck models*?

We thank the reviewer for making this important point. The possibility of selective pressures acting to change heteroplasmy in our models, in addition to the necessity of dealing with samples with different heteroplasmies, is the reason that we (and others) use the normalised heteroplasmy variance V'(*h*) throughout our work. This quantity corrects for the modulation of heteroplasmy variance V(*h*) by mean heteroplasmy E*(h*), and in so doing attempts to remove the effects of selective differences. This protocol has been used in previous work dealing with the NZB/Black mouse system which experience some (small) selective differences (Sharpley et al. Cell 151 333 (2012)). We have now drawn attention to this protocol in the main text. The influence of selective differences on observed heteroplasmy variance in our model can easily be computed by back-transforming the predicted normalised variance using the expected mean heteroplasmy, as can be seen from the structure of the transformation.

*For the new experimental data, the heteroplasmy measurements are reported to a remarkable degree of accuracy. Do the authors really think that they can reliably report down to 0.01% as indicated in the supplementary data*?

In reporting values to this precision we were following convention (e.g. Freyer et al. Nat Genet. 44 1282, 2012). The reviewer is correct in drawing attention to the fact that this reporting could be taken to falsely imply an accuracy of 0.01% in heteroplasmy levels. We have truncated the percentage figures given to 1 d.p. (a more reasonable representation of the data).

*Previous theoretical work has shown that ∼50 heteroplasmy measurements are required to reliably compute heteroplasmy variance. Several of the published datasets do not achieve this, and thus the reported variance measurements are likely to be inaccurate. The same may also apply to the new experimental dataset*.

Figure 4
*raises concerns. For the ∼25 day old data, 6 data points (from a total of 15) are way outside the confidence intervals predicted by the various models. This implies that the new data set is not supporting the current or previous models*.

The reviewer draws attention to the fact that sampling effects may mean that the variance of underlying heteroplasmy distributions is not adequately characterised by previous and current measurements. This is one reason we employ our particular inference protocol, which circumvents problems regarding the characterisation of a “true” heteroplasmy variance. ABC assigns support to a model based on its ability to recapitulate experimental observations, regardless of the interpretation of these observations. Each experimental observation is compared to a simulated outcome involving the same measurement protocol—so, for example, a heteroplasmy variance estimate of 0.01 derived from a sample of 20 cells is compared to 20 instances of stochastic simulation. Whether this estimate of 0.01 represents the “true” heteroplasmy variance does not matter for the inference process—and the distribution of the “true” heteroplasmy variance (corresponding to the variance of the underlying distribution from which samples are taken) is characterised by our approach.

The confidence intervals in Figure 4 do not correspond to the expected spread of individual datapoints, but rather to the spread of the mean variance. An appropriate analogy is to a traditional scatter of datapoints and s.e.m. bars, where many datapoints can lie outside the s.e.m. bars, because the spread of individual datapoints is broader than the spread of possible values for the mean. The fact that several datapoints lie outside the confidence interval for the mean does not therefore represent a weakness of our model. We have endeavoured to make this point clearer in the manuscript by including this discussion in the main text.

*Finally, given the likely readership, it is important to keep reminding readers what is prediction, and what is actual measurement. The section on turnover illustrates this nicely. No turnover measurements have been made; indeed, they would be incredibly difficult to do*.

Throughout our manuscript, experimentally determined data is plotted as circular datapoints and inferred behaviour as lines and shaded regions. We have made this clear in the main text at the point where we describe our first figure, and changed figure captions to explicitly include the fact that predictions have been made.

Reviewer #2:

*There is clearly a great deal of variation in the new data for the normalized variance, but the new data does fit best with the authors' favoured birth-death-partition model. The inclusion of this extra data, at the time points which are most discriminatory with regard to the various models, is therefore helpful, though not absolutely definitive. In particular, it looks as if the spread of the normalised variance itself has interesting dynamics, starting out low, increasing to a maximum at around 25 dpc before diminishing again. Is there any way in which the BDP model can capture these dynamics*?

The apparently rich behaviour of variance is indeed intriguing, but due to technological limitations we cannot discount the possibility that this “variance variance” is due to sampling effects in individual datasets. Without technologically-inaccessible large datasets we cannot confidently characterise higher moments like “variance variance” in any detail and instead focus on “mean variance”. Even this lower-order moment, as discussed above, is subject to sampling issues, so satisfactory experimental characterisation of higher moments is currently impossible. The BDP model does not predict biphasic behaviour of higher moments; but the existing experimental data is not sufficient to characterise any departure between model and experiment as this level of detail.

*Also it was not clear to me why the BDP curve in*
Figure 4
*differs from the HB theory mean plot in*
Figure 4*.*

The difference arises because the trace in Figure 4 arises from a single, best-fit set of model parameters, whereas the trace (and distribution) in Figure 4 describes the behaviour over the inferred posterior range on each parameter. For example, a given parameterisation may yield an onset of increasing V'(*h*) at a particular time (around 40dpc, in the case of the best-fit curve); but different parameterisations may have different values for this onset, “smearing” the increase over a range of times, as seen in the traces for averaged behaviour. We have described the difference between the traces in the main text.

*Finally, I remain concerned that a 2% alteration in mtDNA degradation can have such a large effect, see*
Figure 5*. The authors must address this issue*.

The reviewer raises an important rhetorical point which we have not yet made in a satisfactory way. In Figure 5 we display the effects of two different changes to the model parameterisation. In Figure 5 we change mtDNA degradation and apply a simultaneous and matching change in mtDNA replication. In Figure 5 we are attempting to display the effect of changing mtDNA degradation without a balancing change in mtDNA replication. Our previous attempt to display this effect involved perturbing the difference between degradation and replication rates by 2%. We then, perhaps confusingly, used this as a proxy for mitophagy.

We have, after consideration, used an alternative method. We now change the degradation rate itself by an additive constant, and keep the replication rate constant. This more directly represents the physical degree of freedom we want to address.

The reviewer comments that the magnitude of the effect provoked by a small change in degradation rate is large. This is indeed the case and is related to the distinction between Figure 5 above, and how mtDNA copy number is “controlled”. Copy number control in our simple model is achieved through replication and degradation rates that balance to produce the desired behaviour. The two rates can be varied in concert to maintain this behaviour while changing the timescale of mtDNA turnover, this is what we plot in Figure 5. In Figure 5 we show the effects of perturbing this control by varying the rates independently. The effect of this “unbalanced” perturbation is then indeed pronounced, as we are applying a universal change to the system's expected behaviour. The comparable perturbation to a model applying control based, for example, upon a specific target copy number would be to vary the value of this target.

The selection of specific models for the control of mtDNA populations is a nuanced question and one that we are currently developing further mathematical approaches to address. Preliminary work in this topic has shown that for a range of different, plausibly parameterized, “control strategies” for mtDNA populations (including different incarnations of relaxed replication and setpoint controls, manifest through different birth-death dynamics) the behaviour of heteroplasmy variance is very similar. We therefore believe that our claims regarding stochastic mtDNA turnover likely hold for a number of plausible scenarios, and that further work may shed light on the more specific details of cellular mtDNA control.

We have written more explicitly about these issues in the main text, underlining that our model employs this style of control and that other models (for example, those based on feedback control) could exhibit different behaviours. We will pursue the details of different control mechanisms in further work, but believe that our overall point—that increasing mitophagy may increase V(*h*) whether the system is tightly controlled or not—is valid and illustrated by our work.

[Editors’ note: the author responses to the previous round of peer review follow.]

*Although there was enthusiasm for the quantitative approach to the mitochondrial DNA bottleneck, the reviewers felt that experiments directed at testing the predictions of the modeling are required for the manuscript to significantly advance the field. Therefore, at this stage, we are returning the manuscript to you with an encouragement to submit with experimental data. We hope the reviewers' comments are helpful*.

We have taken your advice and tested predictions of our theoretical approach with an experimental collaboration. We now present an updated version of our study with novel experimental data from an mtDNA model system genetically different to those studied previously in this field. The new data confirms our theoretical predictions and further elucidates the mechanism of the mtDNA bottleneck, while also providing valuable insight about the generality of this process in genetically distinct systems. We believe that this inclusion indeed increases the power of our study, and hope that our improved manuscript will be considered for publication in *eLife*.

We have now, as suggested, collaborated with an experimental group to test the predictions of our theory. Encouragingly, these new experimental measurements demonstrate further support for the model we propose, in addition to demonstrating that these results are consistent across genetically diverse mtDNA pairings. The details of heteroplasmy distributions predicted by our model are also validated by this new data when we train our model on a subset of new datapoints and investigate the fit with the remaining, withheld subset.

On a broad level, Reviewers #1 and #2 were positive about our approach attempting to unify and amalgamate existing experimental data with consistent theory; Reviewer #1 and the editors suggested that confirmatory experimental results would strengthen our findings. We hope that the inclusion of new experimental data confirming our specific theoretical predictions addresses ensures the substance of our scientific contribution and also underlines the powerful cycle of using theoretical approaches both to amalgamate existing experimental data and to propose new experimental strategies to confirm theory and drive further scientific advance. Reviewer #3 raised the criticism that our selection of an existing theoretical model was not novel; we discuss the need for new models, as opposed to novelty in inference and in the analysis of existing models, below, but also hope that the new and confirmatory experimental data addresses some concerns of novelty.

Reviewer #1:

*I do however have some concerns. Principally, the analysis is based entirely on previously published data, which is not all internally consistent. In particular, looking at*
Figure 2
*for the variances, there doesn't seem to be very good agreement between the different datasets (especially comparing the Jenuth data with the rest). I appreciate that the authors' statistical analysis takes into account the experimental uncertainty, but the divergence between the different experimental data sets found in the different studies suggests that the effective experimental error bars may be much bigger than estimated. The ability to distinguish between the models via Bayesian model comparison may then be diminished. This point is important, as there is relatively little difference between the models with respect to the mean values (whose experimental values are also well clustered between the various data sets). Hence it does seem that the variances are the key to distinguishing between the models*.

The reviewer highlights the discrepancies in the source data we use in our inference process. The measured values of heteroplasmy variance in these different studies do indeed cover a wide range. It is worth noting explicitly that these measurements are sample variances and, as the reviewer says, are subject to large uncertainties themselves. We are concerned with the behaviour of the underlying “expected variance”, which the measured datapoints can be envisaged as sampling. Depending on the sampling error involved (demonstrated under one model by Wonnapinij, Chinnery & Samuels (Am J Hum Genet 86:540 [2010])), all the existing data could be viewed as consistent with a given trajectory of this “expected variance”. Most measured variances will lie below this underlying value, as small samples typically do not capture the full variance of a population.

Our modelling approach captures this sampling error by recapitulating the corresponding experimental measurement protocol and comparing the measured outcomes. For example, if a given experimental datapoint reports a variance of 0.01 given 10 individual measurements, we draw 10 samples from the heteroplasmy distribution calculated from a given model and compare the variance of this sample against 0.01. By these means we align our computational simulation with the appropriate experimental protocol. Over the course of the inference and model selection processes, this process identifies those models that are likely to yield sampled variances most comparable to the data.

We have endeavoured to make this picture clearer in the text and in the caption to Figure 3. In addition, we have included new experimental data on heteroplasmy variance, obtained using a consistent experimental protocol that further and independently supports our results. We took these measurements at times chosen to best address the heterogeneous results in existing data and indeed find that the consensus picture from our theoretical treatment amalgamating previous, diverse data, is supported.

*Also the statistical inference is based around a residual sum of squares difference. However, the way this was implemented (using logs of the copy number measurements, for example) seemed a bit arbitrary. Were other possibilities investigated to ensure consistency*?

The reviewer notes that our chosen summary statistic involves some choices, including logs of copy number measurements. Our rationale here is to assign equal importance to matching the observed variability in copy number (which varies over orders of magnitude and is captured by many observations) and the observed variability in heteroplasmy variance (which does not vary over orders of magnitude and is captured by fewer measurements). The protocol employed weights the residuals arising from each of these classes equally, given the number of datapoints available for each. To address the reviewer's concern, we performed our model selection procedure with a range of other protocols, including varying this weighting over several orders of magnitude, and using logarithms of variance measurements as well as copy number. As now described in the manuscript, in all cases, our results held: the BDP model experienced the most statistical support.

*More generally, I am concerned that the modelling is used primarily as a tool to fit existing data and thereby determine which model is most likely on the basis of previously published data. It would be more powerful if the authors could actually test the novel predictions of their modelling in new experiments. This would elevate the study to a higher level. Their model most definitely is predictive (*Table 2*), it's just that they ought to consider teaming up with an experimental group and prove these predictions right*!

The reviewer expresses concern that our modelling approach is primarily a tool to fit existing data and perform model selection given available data, and encourages us to use it predictively. To address this point, we have taken the suggestion of the reviewer and the editors and performed experiments in a new mouse model of mtDNA heteroplasmy involving the pairing of a wild-derived haplotype (labelled HB) with C57BL/6N.

These experiments show that, uninformed by the results in NZB/Balb, the bottleneck in a genetically different system is highly likely to manifest through the same mechanism we identify from previously published data. Moreover, the quantitative details of this mechanism are remarkably comparable between the two genetic systems, with similar amounts of variance increase due to copy number reduction and random turnover. We also test the predictions of our theory regarding the distributional features of heteroplasmy with time by training our model on a subset of these new data and find that our model successfully predicts the variance and distributions of the remaining data.

*One other issue concerns the sensitivity of the system to variations in the mtDNA degradation rate, where even a 2% change causes a huge difference in the mean and variance (*Figure 4*), which is probably unlikely in vivo. This may hint at an underlying weakness of the model*.

The reviewer points out that our model is sensitive to the mtDNA degradation rate. The change in copy number and heteroplasmy variance provoked by a small change in degradation rate is indeed quite high. This theoretical perturbation is perhaps quite a “blunt tool”, corresponding to provoking a deliberate and systematic imbalance of mtDNA production and degradation. We may expect natural mechanisms to compensate for changes in degradation rate by invoking balancing changes in production rate. This is the picture in the previous subfigure, where turnover rather than degradation alone is altered, and the corresponding changes are less dramatic. We include the results for the “unbalanced” changing of mitophagy to aid intuition about how the model system behaves as independent degrees of freedom are varied. We do expect that more homeostatic, ‘mean-reverting' mechanisms are likely to play a role in controlling mtDNA populations over longer timescales; however, the mathematical description of such models is more involved and so we work with this simpler case on the timescale of early development.

Reviewer #2:

*The paper does a good job of summing up the results and conflicting claims of the earlier literature. Their presentation of the differences in the assumptions of the competing models seems accurate to me, and they have built simulation models that encompass the three general competing sets of assumptions. The experimental data from the previous literature is all presented in a combined manner, which is valuable in itself. Through the Beyesian methods of assessing goodness of fit of the combined set of experimental data to the three competing models, the authors present a clear conclusion concerning the model that has the best agreement with the experimental data. This is a valuable contribution to the literature, and again it is very useful that this assessment is coming from an independent set of researchers*.

*While the mathematical detail that comes along with these more sophisticated methods is hard to digest, even for a trained mathematician, I would argue that it is essential to have this level of detail in the publication record*.

The reviewer noted the level of complexity of some of the mathematical analysis involved in the study. The full stochastic process treatment of the birth-death-partitioning model is indeed rather complicated, though powerful; and indeed the derivation of common results used in the literature (such as expected heteroplasmy variance arising from Kimura's neutral model) is also complicated. However, the more straightforward results from this analysis, including expressions for mean heteroplasmy and heteroplasmy variance, are relatively straightforward to write down, and the full treatment affords a powerful way of reasoning about the dynamics of the system.

Reviewer #3:

*There are several statements in the manuscript that are inaccurate and misleading. The notion that organisms ameliorate somatic mutation burden through bottlenecking is a hypothesis (not fact, as written), and not all oocytes have heteroplasmy as stated. Also, the idea that no model has been presented from the ‘bottom up’, with mtDNA as discrete elements, is not correct. The authors cite examples where this was the case (i.e. where the segregating units in the model are proposed to be mtDNA molecules). I also do not accept that there are no methods to allow the comparison of different bottlenecks. One of the authors (Poulton) is a co-author on a Bayesian approach, and Samuels has published extensively on this (some of the papers cited in the current manuscript)*.

The reviewer highlights several misleading statements in the manuscript, specifically pointing out that:

i) The amelioration of mutational damage through the bottleneck is a hypothesis;

ii) Not all oocytes have heteroplasmy;

iii) Discrete models of mtDNA populations do exist;

iv) Statistical analyses of bottleneck mechanisms do exist.

It was not our intention to imply any certainty in points i and ii, and we have altered our Introduction and Abstract to be more compatible with these points. Our claim in point iii is not that no studies exist modelling mtDNA molecules as discrete elements (as the reviewer points out, several studies that we cite take this approach), but rather that no studies yet provide the combination of a broad and general bottom-up model, analytic tractability, explicit time-series modelling of mtDNA population size, and cell divisions. We have changed the appropriate text to make it clearer that we are referring to this combination. We have also altered the text to highlight the fact that statistical analyses of individual bottleneck models and individual datasets have been performed, and added a citation to the Bayesian study that the reviewer cites. We thank the reviewer for helping us clarify the manuscript.

*Some of the proposed mechanisms (which are modelled) have little or no experimental basis. For example, we do not know there is ‘turnover’ or mtDNA within germ cells, and certainly the rate is not known. We have no idea what the size of the nucleoid is in germ cells. There is no evidence that turnover is generally low during cell division to my knowledge, to name but a few. The problem is, by including all of these parameters, the potential outcomes of the modelling will increase*.

The reviewer states that several proposed mechanisms have little or no experimental basis, raising questions including the existence of mtDNA “turnover” and the size of the nucleoid in germ cells. They express concern that including these features may lead to a situation where “potential outcomes of the modelling will increase”, which we take to be a reference to the danger of overfitting—the fact that a suitably complex model can reproduce arbitrarily detailed behaviour but may not represent underlying “truth”.

As our work attempts to take a very general and all-encompassing view of potential bottlenecking mechanisms, the inclusion of a set of features that have been postulated is important. Our Bayesian model selection approach automatically controls against overfitting; this is built into our sampling where the higher the number of parameters, the smaller is the probability that a perturbed set of parameter values will be accepted (analogous to the SMC case in Toni et al. J Roy Soc Interf 6 187 [2008]). As in standard Bayesian model selection, we are effectively integrating over all parameter values, and thus naturally penalising model complexity; unnecessarily complex models will be disfavoured. Furthermore, our parametric inference approach allows us to begin to address some of the uncertainties that the reviewer cites. Part of the message of the paper is that random turnover is the most likely explanation for measured heteroplasmy statistics in later development, and that a quantification of the magnitude of this turnover can be attempted, given data that currently exists. As another example, the reviewer states that we have no idea of nucleoid size in germ cells. But, as our results show, the inheritance of large (>∼ 5) clusters of mtDNAs has little support given existing experimental data, as such inheritance would provoke an increase in heteroplasmy variance rather higher than observed in existing measurements.

Finally, we of course accept (and explicitly state) that the model that we identify as the most supported does not represent unarguable truth. This is the motivation for the series of proposed experiments, by which the individual parameters that we consider can be further elucidated. Our proposed model does encouragingly match the new experimental results we include from a genetically distinct system, but more fine-grained validation of the values of specific parameters should certainly be a target for future experimental work.

*Most importantly, their general conclusions are in keeping with the previous models developed by Samuels, which combine relaxed replication and random segregation contribute to the generation of heteroplasmy in the germ line (i.e. the Cree mechanism, as they describe it). In other words, their findings are confirmatory, not novel*.

The reviewer points out that our general conclusions are in keeping with models developed by Samuels, and argues that our findings are confirmatory rather than novel. An investigative study of existing hypotheses was indeed our intention with this paper, and we argue that unifying and amalgamating existing data to drive new theoretical progress does constitute a novel advance. The problem in the literature was not a shortage of models but an inability to identify the most relevant. We also point out that several of the insights that our work provides (see above) illustrate the power of a quantitative approach in making new theoretical statements, including restricting possible nucleoid sizes and showing the likelihood of mtDNA turnover.

Additionally, our mathematical approach is also a novel analytic solution using generating functions to fully and powerfully capture the stochastic behaviour of a natural system. Reviewers 1 and 2 point out that such a quantitative summary and treatment is valuable; we discuss the issue of novelty further above.

In response to comments from Reviewers 1 and 3 and the editors, we have now included novel experimental data from a genetically distinct system validating our theoretical results and supporting the quantitative predictions that our work makes. We hope that the combination of this new experimental data with the theoretical advances described above addresses concerns of novelty in our study.

*The theoretical manipulation of the bottleneck provides some interesting results, but these are somewhat obvious: if you reduce mtDNA content (by increasing degragadation; incidentally this is something not shown in the germ line), you will accelerate segregation and thus lead to the loss of mutations that reach high heteroplasmy levels. Likewise, it is not surprising that selective pressures influence heteroplasmy variance. Given that heteroplasmy levels are bound by 0% and 100%, any factors pushing the level in a particular direction will inevitably alter the range of values. It is also not surprising that the observed heteroplasmy variance is consistent with a range of bottleneck sizes, given the number of other parameters that can be varied in the described models*.

The reviewer claims that some of our results on manipulating the bottleneck are unsurprising. The qualitative results of some of the perturbations we investigate are indeed intuitive, including reducing mtDNA content to accelerate variance increase. We include these results for several reasons. First, as a quantitative characterisation of the statistically most supported model, illustrating the effect of varying different degrees of freedom of the model. Second, and more importantly, we include these results a quantification of these (perhaps qualitative intuitive) effects, making a numerical prediction regarding, for example, how much mtDNA reduction is required for a given increase in variance. Experimental tests of these predictions form part of our proposed strategies for further elucidation of the model.